# Butterworth as Attention: Anisotropic Spectral Gating for Pansharpening

**Zhenggang Wang** [* 1]   **Wu Wang** [* 1 †]   **Huazhe Liang** [1]   **Tai-Xiang Jiang** [1]

## Abstract

Pansharpening fuses high-resolution panchromatic (PAN) images with low-resolution multispectral (LMS) images. For spatial-spectral fusion, Fast Fourier Transform (FFT)-based methods provide a global receptive field to capture long-range dependencies and naturally separate frequency components. However, most existing approaches directly transplant spatial operators like convolution or self-attention, while disregarding the fundamental structure of the spectrum: a strict spatial correspondence where each coordinate represents a specific frequency component, and a highly non-uniform, radially decaying energy distribution. To address this, we revisit the classical Butterworth filter, a frequency-domain operator defined directly on spectral coordinates that is inherently suited for processing such structured representations. We generalize the standard isotropic Butterworth filter into an anisotropic, learnable frequency-domain gating mechanism, establishing an efficient alternative to self-attention, and propose the Anisotropic Butterworth Fusion Network (ABFNet). Extensive experiments show that ABFNet achieves state-of-the-art (SOTA) performance on pansharpening benchmarks with low computational overhead. Moreover, its strong CIFAR-100 accuracy validates this frequency-domain paradigm's broader applicability. The code is available at https://github.com/Gang-ww/ABFnet.

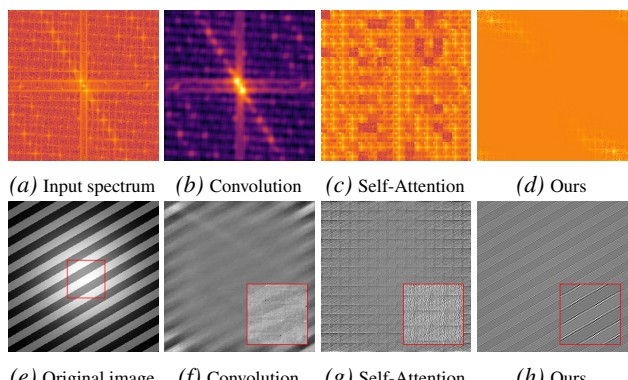

*Figure 1.* Comparison of filtering methods: (a-d) Frequency domain spectra (log scale) showing input spectrum, Convolution filter output, ViT filter output, and our anisotropic Butterworth filter. (e-h) Spatial domain reconstructions showing original image, Convolution result with ringing artifacts, ViT result with grid artifacts, and our clean reconstruction. Red boxes and insets highlight zoom regions.

## 1. Introduction

Pansharpening (Zhang et al., 2023b) aims to generate high-resolution multispectral (HRMS) images by fusing the spatial details from high-resolution panchromatic (PAN) images with the spectral information from low-resolution multispectral (LMS) images. This technique plays a critical role in diverse fields such as environmental monitoring, urban planning, and precision agriculture (Dadrass Javan et al., 2021). Early research primarily relied on traditional signal processing methods, such as Component Substitution (Chavez et al., 1991) and Multi-Resolution Analysis (Vivone et al., 2015). The advent of deep learning has led to significant progress with CNN-based methods (Wang et al., 2025) across various benchmarks. However, CNNs are inherently limited by their local receptive fields, which constrains their ability to model long-range dependencies within an image. Subsequently, Vision Transformers (ViTs) (Huang et al., 2024) achieved superior performance by utilizing self-attention mechanisms to establish a global receptive field. However, this capability comes at a substantial cost. Not only do the core matrix operations in self-attention introduce a computational complexity of O(N²), leading to prohibitive overhead, but the self-attention mechanism also tends to act as a low-pass filter (Wang et al., 2022). This inclination toward suppressing high-frequency information is particularly detrimental for Pansharpening, a task in which preserving and enhancing high-frequency details is crucial.

---
[*]Equal contribution [1]School of Computing and Artificial Intelligence, Southwestern University of Finance and Economics, Chengdu, Sichuan, China. Correspondence to: Wang Wu <wangwu@smail.swufe.edu.cn>.

*Proceedings of the 43rd International Conference on Machine Learning*, Seoul, South Korea. PMLR 306, 2026. Copyright 2026 by the author(s).

In parallel, FFT has attracted growing attention not only for its ability to provide a global receptive field with efficient $O(N \log N)$ computational complexity (Rao et al., 2021), but also for its intrinsic property of operating in the frequency domain. This allows signals to be naturally decomposed into different frequency bands, enabling targeted processing and better preservation of crucial high-frequency information. However, existing frequency-domain methods often directly transplant operators designed for the spatial domain, such as convolution (Hou et al., 2023; Liu et al., 2025; Luo et al., 2025) or self-attention (Zhang et al., 2023a). This approach overlooks a fundamental discrepancy in information distribution: spatial images have translation invariance, whereas different positions in a frequency spectrum correspond to distinct types of information. Specifically, the center of the spectrum corresponds to low-frequency components representing smooth contours, and the periphery corresponds to high-frequency components capturing fine textures. Furthermore, spectral energy follows a highly non-uniform distribution, being concentrated at the center. This intrinsic difference necessitates operators specifically designed for the spectral domain.

To concretely illustrate the negative effects of directly applying spatial operators to the frequency domain, we constructed a toy dataset containing oriented high-frequency textures and simulated two representative processing mechanisms, As shown in Fig. 1: simulating CNN-style translation invariance by applying a fixed $9 \times 9$ sliding-window average filter on the logarithmic magnitude spectrum, which resulted in severe distortion in the reconstructed spatial image; and simulating ViT-style patch splitting by partitioning the continuous spectrum into non-overlapping $16 \times 16$ patches and computing self-attention within each patch, which destroyed spectral continuity and introduced unnatural grid artifacts upon reconstruction.

These experiments confirm that naively transplanting spatial operators damages the structural integrity of spectral data. Consequently, an operator native to the frequency domain is essential. The Butterworth filter, characterized by a flat passband in its frequency response, offers a natural and effective foundation (as supported by comparative experiments using a Gaussian filter in the FFT domain, where the Butterworth filter yields superior performance; see Sect. 5.7). More importantly, its mathematical formulation can be interpreted as a specialized form of self-attention that inherently possesses a global receptive field. Specifically, the learnable cutoff frequency and filter order function analogously to the Query and Key in a self-attention, while a term containing frequency coordinates acts as a dynamic positional encoding. Their interaction produces an attention-like distribution, enabling adaptive weighting across frequency bands. Crucially, unlike standard self-attention that relies on Softmax normalization and expensive matrix multipli-

cation, this Butterworth-based operator employs a Sigmoid function to generate a gating mask. This mask performs element-wise multiplication with the input spectral features (serving as the Value), preserving the global receptive field while reducing computational complexity to $O(N \log N)$. For a schematic comparison, see Fig. 2 and Sect. 3.2.

Inspired by these insights, we propose ABFNet, a novel pansharpening framework built upon a learnable, anisotropic Butterworth filter as its core operator. The network is constructed by stacking multiple Spectral Fusion Blocks (SFBs), each following the structure of a standard Transformer block that consists of a self-attention layer and a feed-forward network (FFN). A key innovation of our design is the replacement of the conventional self-attention layer with our proposed Butterworth Attention Module (BAM). The BAM implements a dual-branch gating mechanism in the frequency domain, where both branches perform anisotropic Butterworth filtering: the global branch processes the entire feature map to capture holistic context, while the local branch employs a two-stage strategy to model regional variations. Specifically, within the local branch, Butterworth Token Interaction (BTI) first establishes patch-wise dependencies through efficient frequency-domain communication; these dependencies then guide a second stage of patch-specific anisotropic filtering to adaptively enhance directional textures. By stacking multiple SFBs, ABFNet achieves a global receptive field with low computational overhead, offering an efficient and effective paradigm for remote sensing image fusion.

In summary, the main contributions of this paper are as follows:

- We revisit and generalize the classical Butterworth filter into a learnable, anisotropic gating module that operates directly in the frequency domain. This provides a structurally coherent and computationally efficient core operator for global context modeling.

- Built upon this operator, we introduce the Butterworth Token Interaction, a novel mechanism that arranges patch-wise filter parameters into a 2D map and performs global interaction via frequency-domain filtering. This elegantly replaces the O(N²) pairwise computations of standard self-attention with an O(N log N) process.

- We integrate these designs into a dual-branch network, ABFNet, which effectively balances holistic spectral modulation with locally adaptive, direction-aware filtering for remote sensing image fusion.

- Extensive experiments demonstrate that ABFNet achieves state-of-the-art performance on pansharpening benchmarks with low computational cost. The

strong generalization ability of its core module is further validated on the CIFAR-100 classification task, indicating applicability beyond remote sensing.

CONFLICT OF INTEREST DISCLOSURE

The authors declare no conflicts of interest.

## 2. Related Work

### 2.1. Frequency Domain Pansharpening

Pansharpening in the frequency domain enables multi-scale feature separation and global modeling at a lower computational cost. Prior work includes multi-resolution transforms, such as pyramid-based (He et al., 2023) and wavelet-based (Huang et al., 2025) methods.

Recent FFT-based methods form a distinct category. These approaches typically employ dual-stream structures to couple spatial features with amplitude and phase spectra (Zhou et al., 2022), reconstruct amplitude and phase separately via convolution (Hou et al., 2023), or perform explicit spatial-frequency aggregation (Liu et al., 2025). A central limitation of these FFT-based methods is their reliance on spatial operators (e.g., convolution or self-attention) applied directly to the complex spectrum. This design fails to fully respect the intrinsic structural correspondence between spatial coordinates and frequencies, often leading to suboptimal representation and redundant computation.

### 2.2. Frequency Learning in General Vision

Beyond pansharpening, frequency domain learning has thrived in general computer vision, offering efficient global feature extraction and long-range dependency modeling. Frequency-domain methods in general vision fall into three paradigms. **(a) Spatial transplantation** directly applies convolution or attention to spectral features: Fast Fourier Convolution (Chi et al., 2020) leverages the global receptive field for inpainting, FcaNet (Qin et al., 2021) innovates channel attention via multi-spectral pooling, and FSANet (Zhang et al., 2023a) reformulates self-attention in the frequency domain for dense prediction. As analyzed in Sect. 3, however, such transplantation ignores the structured coordinate semantics of the spectrum and reduces to coordinate-dependent point-wise scaling rather than true spatial aggregation. **(b) Unconstrained global filters** instead learn element-wise weights directly on the frequency grid. GFNet (Rao et al., 2021) and SpectFormer (Patro & Agneeswaran, 2023) exemplify this paradigm as efficient alternatives to self-attention, with subsequent works extending it to deepfake detection (Tan et al., 2024), camouflaged object detection (Deng et al., 2025), and dense prediction (Chen et al., 2025). A shared limitation is that filter weights are tied to a fixed resolution grid, causing discretiza-

tion mismatch across scales. **(c) Coordinate-transformed attention** remaps the spectrum to an alternative system (e.g., polar coordinates) before applying attention, aligning the operator with the radially decaying energy distribution of natural spectra. While conceptually appealing, coordinate resampling inevitably introduces spectral leakage and geometric distortion, undermining the Cartesian structural integrity essential for stable optimization.

## 3. Background and Key Observation

### 3.1. Butterworth Filter Revisited

The Butterworth filter is a classical low-pass filter defined in the FFT domain. Its standard operational pipeline comprises three stages: (1) transforming a spatial-domain image into the frequency domain via the FFT to obtain its spectral representation; (2) modulating this spectrum through element-wise multiplication with a filter transfer function; and (3) mapping the processed spectrum back to the spatial domain via the Inverse FFT (IFFT). The classical Butterworth filter achieves smooth, global control over the image spectrum by adjusting only two scalar parameters: the cut-off frequency $D_0$ and the filter order $n$. This parametric simplicity renders it an ideal candidate for integration into differentiable deep learning frameworks.

For a 2D image, the transfer function of a classic $n$-th order low-pass Butterworth filter, which acts as the modulation mask, is defined as:

$$H(u,v) = \frac{1}{1 + \left(\frac{D(u,v)}{D_0}\right)^{2n}}, \qquad (1)$$

where $(u,v)$ denotes the frequency-domain coordinates, and $D(u,v) = \sqrt{u^2 + v^2}$ is the Euclidean distance from point $(u,v)$ to the spectrum center (the DC component), physically representing the radial frequency magnitude. The filter is applied via an element-wise multiplication with the complex image spectrum.

### 3.2. Key Observation: Butterworth Filtering as Self-Attention

To integrate this classical mechanism into deep neural networks, we first reformulate its mathematical expression. Applying the identity $x = e^{\ln x}$ to the core term in (1) yields:

$$\left(\frac{D(u,v)}{D_0}\right)^{2n} = \exp\Big(2n \cdot \big(\ln D(u,v) - \ln D_0\big)\Big). \quad (2)$$

Substituting (2) back into (1) leads to a form directly recognizable as a Sigmoid function, $\sigma(z) = (1 + e^{-z})^{-1}$:

$$H(u,v) = \frac{1}{1 + \exp\Big(2n \cdot \big(\ln D(u,v) - \ln D_0\big)\Big)}. \quad (3)$$

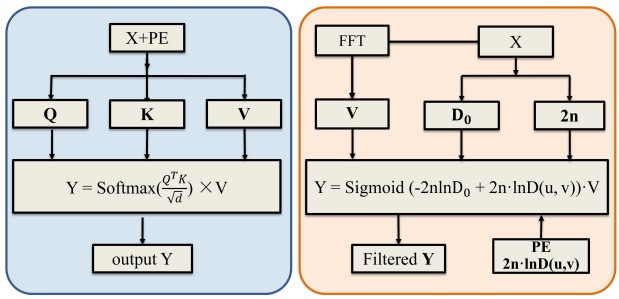

*Figure 2.* Architectural comparison between Standard Self-Attention and our proposed Frequency-Domain Butterworth Attention. We reformulate $Q$, $K$, and Positional Encoding (PE) into parameters $D_0$, $2n$, and $2n \cdot \ln D(u,v)$. This design replaces the $O(N^2)$ matrix multiplication with $O(N \log N)$ element-wise gating via a Sigmoid mask $H(u,v)$.

Therefore, we ultimately derive the formula:

$$H(u,v) = \sigma\left(-2n \cdot \ln D_0 + 2n \cdot \ln D(u,v)\right). \quad (4)$$

This reformulation reveals a profound connection to the self-attention mechanism. We observe that the derived expression in (3) functions analogously to a frequency-domain gating mechanism: The logarithmic cutoff frequency, $D_0$, acts as the Query (Q) by defining the target frequency band, while the filter order, $2n$, serves as the Key (K), controlling the slope of the response and thus the model's selectivity or sensitivity to frequency deviations. By appropriately adjusting the sign, we ensure that the formulation aligns with the standard low-pass Butterworth filter expectation: passing low frequencies with a flat response while smoothly attenuating higher frequencies beyond the cutoff. Furthermore, the term $2n \cdot \ln D(u,v)$, which incorporates the inherent frequency coordinates $(u,v)$, functions as a Dynamic Positional Encoding. It allows the model to adaptively assign weights based on an individual frequency component's location in the spectrum. A visual comparison of their corresponding workflows is provided in Fig. 2.

The fundamental distinction from standard self-attention lies in the interaction mechanism. While self-attention computes a dense affinity matrix $A = \text{Softmax}(QK^T/\sqrt{d_k})$ followed by a weighted sum $A \times V$, the Butterworth filter employs a Sigmoid activation to generate a gating mask $H$. The input spectral features themselves act as the Value (V). The final output is obtained via an efficient element-wise product as follows:

$$Y(u,v) = H(u,v) \odot V(u,v). \quad (5)$$

This design avoids the expensive $O(N^2)$ computation of self-attention. The core complexity is reduced to $O(N \log N)$ for element-wise gating, while retaining a global receptive field via the FFT. This establishes the Butterworth filter as an efficient alternative for global context

modeling in the frequency domain. We further demonstrate the Butterworth filter is functionally equivalent to a form of self-attention whose attention mask exhibits a specific, mathematically structured pattern in Supplementary Material S1 and S3.

### 3.3. From Isotropic to Anisotropic

The standard Butterworth filter, as formulated in Sect. 3.2, is isotropic with a circularly symmetric frequency response. This is a limitation for pansharpening, where high-frequency details (e.g., urban grids, field boundaries) often exhibit strong directional patterns. To adapt to such anisotropic structures, we generalize the filter to an orientation-aware form. We achieve this by projecting the original frequency coordinates $(u,v)$ into a rotated system $(u',v')$ via a rotation matrix defined by an angle $\theta$:

$$\begin{bmatrix} u' \\ v' \end{bmatrix} = \begin{bmatrix} \cos\theta & \sin\theta \\ -\sin\theta & \cos\theta \end{bmatrix} \begin{bmatrix} u \\ v \end{bmatrix}. \quad (6)$$

In the rotated coordinate system, the Euclidean distance $D(u,v)$ is decomposed into two axial components, each processed by separate cutoff frequencies $D_{u0}$ and $D_{v0}$. This transforms the originally circular passband into an elliptical one, enabling the model to selectively emphasize textures along specific orientations. The Anisotropic Elliptical Ratio $\mathcal{R}(u_0', v_0')$ is defined as:

$$\mathcal{R}(u',v') = \sqrt{\left(\frac{u'}{D_{u0}}\right)^2 + \left(\frac{v'}{D_{v0}}\right)^2}. \quad (7)$$

Substituting this ratio into (4) yields the final Anisotropic Butterworth transfer function $H(u,v)$:

$$H(u,v) = \sigma\left(-2n \cdot \ln \mathcal{R}(u',v')\right). \quad (8)$$

## 4. Methodology

Building upon the mathematical correspondence between the Butterworth filter and self-attention established in Sect. 3, we introduce the Anisotropic Butterworth Fusion Network (ABFNet). As illustrated in Fig. 3, ABFNet adopts a streamlined pipeline for pansharpening: it takes a LMS image $I_{\text{LMS}}$ and a PAN image $I_{\text{PAN}}$ as inputs. After concatenation and initial feature extraction via a $3 \times 3$ convolution, a sequence of $N$ SFBs performs deep feature integration. Finally, a reconstruction layer produces the high-resolution fused output $I_{\text{Fused}}$.

### 4.1. Spectral Fusion Block (SFB)

Each SFB follows a Transformer-like hierarchical structure, with its core component being the Butterworth Attention Module (BAM) which replaces conventional self-attention. To further enhance local spatial representation, we employ

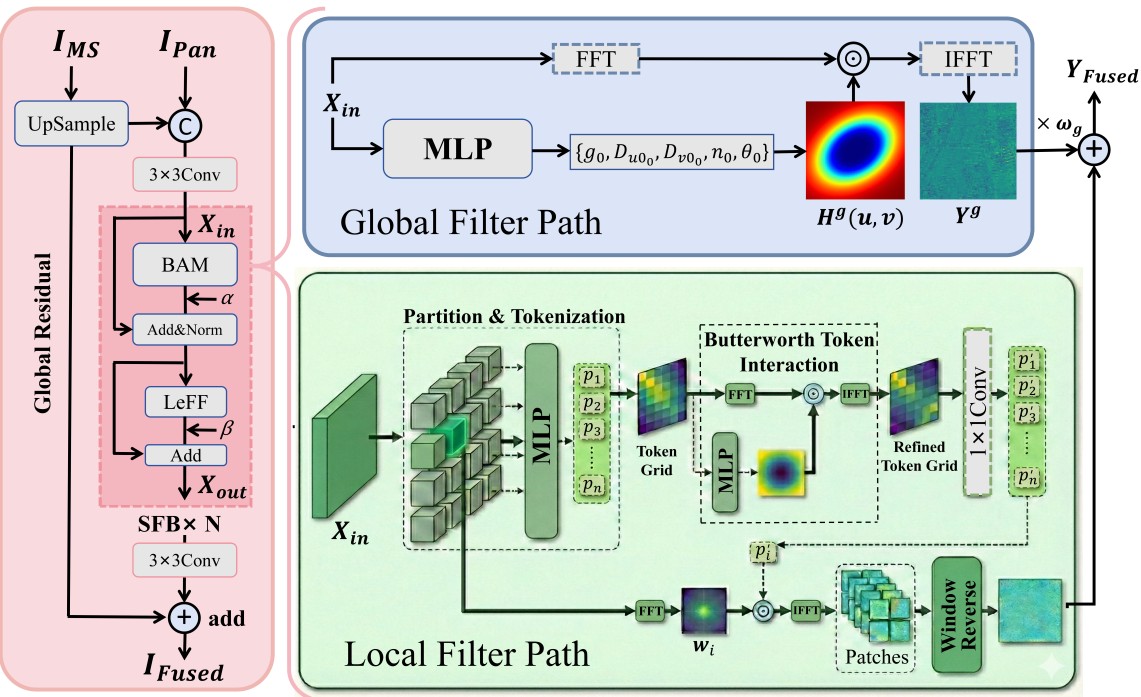

*Figure 3.* **Overview of the proposed ABFNet architecture.** The left panel illustrates the overall network framework, while the right panel details the Butterworth Attention Module (BAM), which comprises a global filter path (top) and a local filter path (bottom).

a Locally-enhanced Feed-Forward Network (LeFF) (Yuan et al., 2021) in place of the standard FFN. Formally, for an input feature $X_{in} \in \mathbb{R}^{B \times C \times H \times W}$, the SFB output $X_{out}$ is:

$$X_{mid} = \text{LayerNorm}\left(X_{in} + \alpha \cdot \text{BAM}(X_{in})\right),$$
$$X_{out} = X_{mid} + \beta \cdot \text{LeFF}(X_{mid}), \tag{9}$$

where $\alpha$ and $\beta$ are learnable scaling parameters.

### 4.2. Butterworth Attention Module (BAM)

Building upon the Butterworth-as-Attention reformulation, we design the BAM to replace standard self-attention. To simultaneously capture holistic image context and local directional textures, BAM employs a complementary dual-branch architecture: a Global Filtering Path and a Local Filtering Path, both of which implement the anisotropic Butterworth filtering in the frequency domain.

#### 4.2.1. PARAMETER GENERATION AND MASK CONSTRUCTION

Following the principle that self-attention dynamically generates Q, K, and V matrices from input tokens, BAM learns to generate five specific filtering parameters from the input features: $\{D_{u0}, D_{v0}, \theta, n, g\}$. The first four define the core anisotropic Butterworth filter $H(u, v)$ as in (7) and (8). Unlike the conventional low-pass Butterworth filter, which is suboptimal for pansharpening where high-frequency detail

must be enhanced, we introduce a trainable gating parameter $g$ constrained to $[0, 1]$ to adaptively modulate the spectral response. The final frequency-domain gating mask is then given by:

$$B(u, v) = g \cdot (1 - H(u, v)) + (1 - g) \cdot H(u, v). \tag{10}$$

This formulation allows $g$ to dynamically control the trade-off between low-pass and high-pass behavior, enabling the module to preserve or accentuate frequency components as required by the task.

#### 4.2.2. GLOBAL FILTERING PATH

The Global Path instantiates the above mechanism at the full-image scale to model holistic structure. A lightweight sub-network (GAP + two-layer MLP) extracts global statistics to generate a single parameter set $(\theta^g, D_{u0}^g, D_{v0}^g, n^g, g^g)$, which defines a global anisotropic mask $B^g(u, v)$. The output is obtained by modulating the spectrum of the input feature map $\mathbf{X}$:

$$Y^g = \text{IFFT}\left(\text{FFT}(\mathbf{X}) \odot B^g(u, v)\right). \tag{11}$$

As shown in Fig. 4, the estimated dominant texture orientation (red line in column 1) aligns closely with the principal smoothing direction of the learned filter in the spatial domain (red dashed line in column 5). This alignment demonstrates that the global path successfully learns anisotropic Butterworth filters that adaptively enhance details along the dominant texture directions.

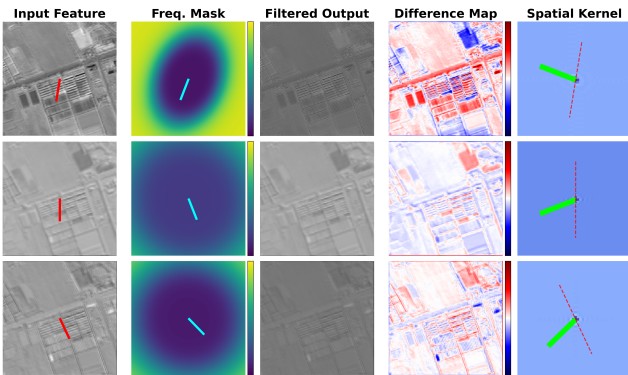

*Figure 4.* Visualization of the learned anisotropic filtering process on selected feature channels. From left to right: the input feature map with its dominant orientation (red line) estimated via the structure tensor (Bigün et al., 1991); the corresponding anisotropic Butterworth frequency mask in the Fourier domain, whose passband major axis (cyan arrow) aligns with the estimated direction; the filtered output obtained via inverse FFT; the difference map $Y^g - X_{\text{in}}$, highlighting selectively enhanced or suppressed frequency components; and the equivalent spatial-domain convolution kernel (contrast-adjusted for clarity), whose principal smoothing direction (green arrow) matches the input texture orientation (red dashed line).

#### 4.2.3. LOCAL FILTERING PATH AND BUTTERWORTH TOKEN INTERACTION (BTI)

The local path adapts the global Butterworth filtering mechanism to the patch level, enabling the modeling of regional variations and directional textures. Its core is the BTI module, which facilitates efficient global communication between patches through a condensed latent representation. The process consists of four stages: patch tokenization, spectral interaction, parameter decoding, and patch-wise filtering.

**Patch Partitioning and Latent Tokenization.** The input feature map $\mathbf{X}_{in}$ is partitioned into $N$ non-overlapping patches, each denoted as $\mathbf{X}_i \in \mathbb{R}^{C \times M \times M}$ for $i = 1, \ldots, N$. A shared lightweight predictor (patch-wise global average pooling followed by an MLP) condenses each patch $\mathbf{X}_i$ into a latent token $\mathbf{P}_i \in \mathbb{R}^{C \times 1 \times 1}$, serving as its semantic fingerprint. We denote the set of tokens as $\mathcal{P} = \{\mathbf{P}_1, \ldots, \mathbf{P}_N\}$.

**Anisotropic Spectral Interaction.** The token set $\mathcal{P}$ is rearranged into a macroscopic grid $\mathbf{P}_{\text{grid}} \in \mathbb{R}^{B \times C \times H_p \times W_p}$ (where $H_p = H/M, W_p = W/M$) to preserve spatial relationships. Global interaction is then achieved by applying an Anisotropic Butterworth Filter in the frequency domain:

$$\mathbf{P}_{\text{refined}} = \text{IFFT}\left(\text{FFT}(\mathbf{P}_{\text{grid}}) \odot \mathcal{H}_{\text{grid}}(u, v)\right), \quad (12)$$

where $\mathcal{H}_{\text{grid}}$ is the learnable spectral mask, allowing each token to aggregate context from all others under directional guidance.

**Parameter Decoding and Modulation.** The refined tokens

$\mathbf{P}_{\text{refined}}$ are decoded by a $1 \times 1$ convolution to generate the final per-patch filtering parameters $\{D_{u0}^i, D_{v0}^i, \theta^i, n^i, g^i\}$ for $i = 1, \ldots, N$.

**Patch-Wise Filtering and Aggregation.** Each original patch $\mathbf{X}_i$ is independently filtered in the frequency domain using its corresponding anisotropic Butterworth mask $H_i^l(u, v)$ constructed from the decoded parameters:

$$\mathbf{y}_i = \text{IFFT}\left(\text{FFT}(\mathbf{X}_i) \odot H_i^l(u, v)\right). \quad (13)$$

The filtered patches $\{\mathbf{y}_1, \ldots, \mathbf{y}_N\}$ are then reassembled to form the local output $Y^l$. The output features $Y^g$ and $Y^l$ are adaptively fused via a learnable weight $w_g$ to obtain $F_{\text{fused}}$:

$$F_{\text{fused}} = w_g \cdot Y^g + Y^l. \quad (14)$$

### 4.3. Loss Function

To guide the network in generating fused images with sharp spatial details and spectral fidelity, we adopt the $L_1$ norm as the reconstruction loss, directly minimizing the pixel-wise absolute error between the network prediction $I_{\text{Fused}}$ and the ground truth high-resolution multispectral image $I_{\text{GT}}$:

$$\mathcal{L}_{\text{rec}} = \frac{1}{T} \sum_{i=1}^{T} \left\| I_{\text{Fused}}^{(i)} - I_{\text{GT}}^{(i)} \right\|_1. \quad (15)$$

## 5. Experiments

### 5.1. Datasets and Evaluation Metrics

To comprehensively evaluate the generalization ability of the model, we selected three widely used remote sensing satellite datasets: WorldView-3 (WV3), Gaofen-2 (GF2), and QuickBird (QB). For each dataset, we followed the Wald protocol (Wald, 2000) to construct the reduced-resolution training and testing sets with a reference image. To assess cross-satellite generalization, we additionally evaluate zero-shot transfer to WorldView-2 (WV2), using models trained on WV3 and tested without any fine-tuning. Regarding the evaluation framework, we employed two complementary sets of metrics. For reduced-resolution and zero-shot experiments, we used Peak Signal-to-Noise Ratio (PSNR), Spectral Angle Mapper (SAM) (Yuhas et al., 1992; Kruse et al., 1993), and Erreur Relative Globale Adimensionnelle de Synthèse (ERGAS) (Wald, 2000). For full-resolution experiments, we adopted the no-reference quality indices including the spectral distortion index $D_\lambda$, the spatial distortion index $D_s$, and the hybrid quality metric HQNR (Alparone et al., 2008). For more implementation details, please refer to the supplementary materials.

### 5.2. Comparison with SOTAs

We conduct a comparison with eight current state-of-the-art pansharpening methods, including ARConv (Wang et al.,

*Table 1.* Average metrics on reduced-resolution WV3, QB, and GF2 samples, and zero-shot cross-sensor transfer on WV2. Best results are bolded, second-best underlined.

| Method | WV3 Reduce | | | QB Reduce | | | GF2 Reduce | | | WV2 Zero-Shot | | | Params (M)↓ | FLOPs (G)↓ |
|---|---|---|---|---|---|---|---|---|---|---|---|---|---|---|
| | PSNR↑ | ERGAS↓ | SAM↓ | PSNR↑ | ERGAS↓ | SAM↓ | PSNR↑ | ERGAS↓ | SAM↓ | PSNR↑ | SAM↓ | ERGAS↓ | | |
| SFIIN (Zhou et al., 2022) | 34.29 | 3.01 | 4.31 | 36.00 | 4.74 | 5.49 | 47.32 | 1.03 | 1.21 | 28.04 | 6.01 | 4.86 | 0.09 | 5.37 |
| TDNet (Zhang et al., 2023c) | 36.03 | 2.54 | 3.64 | 37.17 | 4.15 | 4.58 | 47.27 | 1.02 | 1.23 | 26.68 | 6.04 | 9.23 | 0.49 | 20.00 |
| BIMPan (Hou et al., 2023) | 36.95 | 2.30 | 3.44 | 36.14 | 4.67 | 4.47 | 45.92 | 1.17 | 1.35 | **29.58** | 5.43 | **4.03** | 2.34 | 29.45 |
| DISPnet (Wang et al., 2024a) | 36.80 | 2.33 | 3.42 | 36.83 | 4.33 | 4.47 | 48.76 | 0.86 | 1.02 | 28.46 | 5.81 | 4.61 | 0.33 | 25.60 |
| DCINN (Wang et al., 2024b) | 37.22 | 2.24 | 3.30 | 38.43 | 3.57 | 4.22 | 49.28 | 0.81 | 0.98 | 27.93 | 5.82 | 4.94 | 0.49 | 13.62 |
| WFAnet (Huang et al., 2025) | 36.60 | 2.39 | 3.46 | 38.48 | 3.54 | 4.63 | 48.56 | 0.88 | 1.05 | 28.08 | 6.06 | 4.87 | 0.53 | 14.33 |
| ARConv (Wang et al., 2025) | 36.58 | 2.38 | 3.41 | 37.67 | 3.90 | 4.99 | 45.45 | 1.23 | 1.43 | 28.61 | 5.58 | 4.51 | 4.79 | 61.57 |
| BNNPan (Hou et al., 2025) | 36.00 | 2.54 | 3.63 | 37.66 | 3.90 | 5.07 | 40.77 | 1.09 | 1.26 | 27.43 | 6.47 | 5.16 | 0.25 | 15.28 |
| **Ours Only Global** | 36.71 | 2.36 | 3.40 | 38.36 | 3.59 | 4.19 | 48.78 | 0.85 | 1.00 | -- | -- | -- | **0.07** | **3.58** |
| **Ours Full** | **37.65** | **2.13** | **3.17** | **38.83** | **3.41** | **4.06** | **50.30** | **0.72** | **0.86** | 29.05 | **5.37** | 4.29 | 0.26 | 12.64 |

*Table 2.* Full-resolution evaluation on WV3, QB, and GF2 datasets using no-reference quality indices, with empirical inference latency and peak GPU memory measured on a single NVIDIA RTX 5060 Ti (batch size 1, FP32, $64\times64$ LMS / $256\times256$ PAN). Best results are bolded, second-best underlined. OOM = out of memory.

| Method | WV3 Full | | | QB Full | | | GF2 Full | | | Latency (ms)↓ | Peak Mem. (MB)↓ |
|---|---|---|---|---|---|---|---|---|---|---|---|
| | HQNR↑ | $D_\lambda$↓ | $D_S$↓ | HQNR↑ | $D_\lambda$↓ | $D_S$↓ | HQNR↑ | $D_\lambda$↓ | $D_S$↓ | | |
| SFIIN (Zhou et al., 2022) | 0.838 | 0.120 | 0.051 | 0.762 | 0.185 | 0.069 | 0.750 | 0.101 | 0.060 | 9.29 | 76.03 |
| TDNet (Zhang et al., 2023c) | 0.925 | 0.032 | 0.045 | 0.880 | 0.080 | 0.044 | 0.916 | 0.029 | 0.050 | **6.90** | 84.20 |
| BIMPan (Hou et al., 2023) | 0.941 | 0.029 | 0.032 | 0.883 | 0.067 | 0.055 | 0.789 | 0.038 | 0.076 | 192.75 | 342.23 |
| DISPnet (Wang et al., 2024a) | 0.947 | 0.027 | 0.027 | 0.892 | 0.073 | 0.037 | 0.755 | 0.207 | 0.048 | 29.70 | 101.90 |
| DCINN (Wang et al., 2024b) | **0.954** | 0.024 | **0.023** | 0.898 | **0.036** | 0.069 | 0.920 | 0.029 | 0.053 | 10.27 | **56.15** |
| WFAnet (Huang et al., 2025) | 0.950 | 0.027 | 0.024 | 0.855 | 0.054 | 0.096 | 0.837 | 0.030 | **0.034** | OOM | OOM |
| ARConv (Wang et al., 2025) | 0.953 | **0.023** | 0.025 | **0.915** | 0.052 | **0.035** | 0.842 | 0.066 | 0.100 | 92.56 | 671.58 |
| BNNPan (Hou et al., 2025) | 0.928 | 0.036 | 0.038 | 0.847 | 0.094 | 0.067 | 0.811 | **0.028** | 0.063 | 31.60 | 159.18 |
| **Ours Full** | 0.951 | **0.023** | 0.028 | 0.891 | 0.050 | 0.063 | **0.937** | **0.028** | 0.037 | 38.19 | 297.89 |

2025), WFAnet (Huang et al., 2025), BNNPan (Hou et al., 2025), DCINN (Wang et al., 2024b), BIMPan (Hou et al., 2023), TDnet (Zhang et al., 2023c), DISPnet (Wang et al., 2024a), and SFIIN (Zhou et al., 2022). Notably, SFIIN and BIMPan, similar to our proposed ABFNet, are representative approaches that leverage the FFT domain for pansharpening.

**Quantitative Evaluation.** As reported in Table 1, ABFNet achieves SOTA performance on all reduced-resolution benchmarks, with PSNR gains of 0.43–0.97 dB over the best competing methods on WV3, QB, and GF2, while also leading in ERGAS and SAM. In zero-shot cross-sensor transfer to WV2 (trained solely on WV3), ABFNet achieves the best SAM of 5.37 and the second-best PSNR and ERGAS, simultaneously outperforming six of eight competitors across all three metrics and demonstrating strong cross-satellite generalization. The Only Global variant further reduces overhead to 0.07M parameters and 3.58G FLOPs while remaining competitive. In full-resolution evaluation (Table 2), ABFNet achieves the best HQNR of 0.937 and lowest $D_\lambda$ on GF2, while remaining competitive on WV3 and QB. Regarding runtime efficiency, methods relying purely on convolution (e.g., SFIIN, TDNet) achieve lower latency via optimized CUDA kernels but sacrifice long-range modeling capacity. Among methods with a global receptive field, ABFNet achieves the best efficiency–capability trade-off: WFANet runs out of memory, and ARConv consumes 2.4× our latency and 2.3× our peak GPU memory.

**Qualitative Visualization.** Visual comparisons further substantiate these findings. As illustrated in Fig. 5 and Fig. 6, ABFNet produces sharper building outlines and clearer textures, with residual maps showing the lowest error intensity. On full-resolution samples, our method excels in spectral preservation, maintaining natural green hues in vegetation areas where other methods suffer from unnatural saturation or distortion.

*Table 3.* Ablation study of ABFNet components. The best results are highlighted in **bold**.

| Variant | WV3 | QB | | GF2 |
|---|---|---|---|---|
| | PSNR↑ | PSNR↑ | HQNR↑ | PSNR↑ |
| Only Global | 36.72 | 38.36 | 0.877 | 48.78 |
| Only Local | 37.62 | 38.75 | 0.873 | 50.13 |
| Isotropic | 37.62 | 38.71 | 0.883 | 49.99 |
| No BTI | 37.62 | 38.71 | 0.884 | 50.20 |
| **Full Model** | **37.65** | **38.83** | **0.891** | **50.30** |

### 5.3. Ablation Studies

We evaluate the effectiveness of the core components and the physical design rationale of ABFNet through four variant models (Table 3): 1) **Only Global**, 2) **Only Local**, 3) **Isotropic**, and 4) **No BTI**. The results demonstrate that the Full Model achieves superior performance across all benchmarks by parallelly integrating multi-scale frequency features. Specifically, Only Local generally outperforms

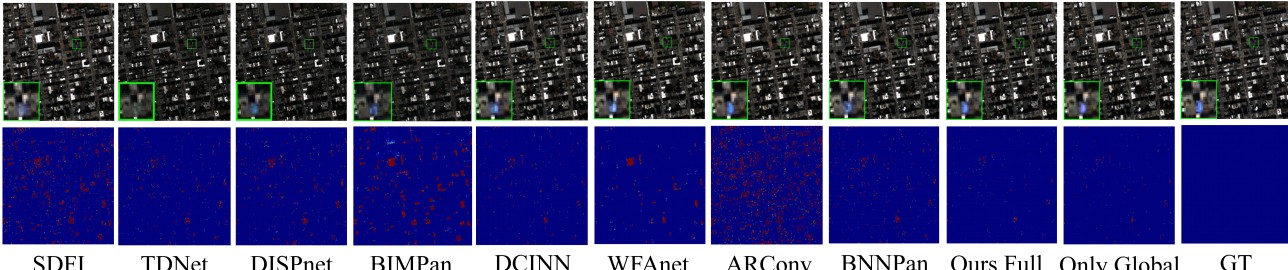

SDFI  TDNet  DISPnet  BIMPan  DCINN  WFAnet  ARConv  BNNPan  Ours Full  Only Global  GT

*Figure 5.* Qualitative visual comparison (Top) and corresponding absolute error maps (Bottom) for various methods on the QB dataset. The red regions in the residual maps indicate higher errors compared to the Ground Truth (GT).

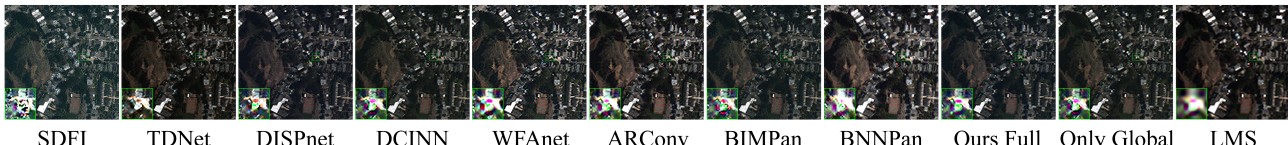

SDFI  TDNet  DISPnet  DCINN  WFAnet  ARConv  BIMPan  BNNPan  Ours Full  Only Global  LMS

*Figure 6.* Visual comparison of different pansharpening methods on a full-resolution sample. Compared with other SOTA methods, our results exhibit the highest spectral fidelity and preserves the most natural hues, demonstrating superior color consistency with real-world scenarios.

Only Global, confirming that fine-grained spectral characteristics in local windows are more critical for detail restoration. Furthermore, the performance gain over the Isotropic variant validates that our anisotropic design is better suited for capturing the directional high-frequency textures inherent in remote sensing imagery. Moreover, we observe that removing the BTI module leads to a consistent and comprehensive degradation across all evaluation metrics. This confirms that enabling implicit, frequency-domain interaction among latent parameter tokens is crucial for coordinating spatially-adaptive filtering across the image.

### 5.4. Branch Fusion Strategy

To justify our per-channel learnable weighting for combining the global and local branch outputs, we compare it against four alternatives on GF2 (Table 4): **Average** (fixed equal-weight sum), **Cross-Attention** (key-query dynamic weights), **Concat** (concatenation + $1\times1$ conv), and $w_g$ **Global** (single global scalar). Cross-Attention incurs +4.42G FLOPs and +0.08M parameters while underperforming. Concat improves quality marginally at extra cost. $w_g$ Global matches our budget but loses channel-wise frequency adaptivity. Our design achieves the best efficiency-adaptivity trade-off.

### 5.5. Operator Analysis in Frequency Domain

To further investigate the impact of different operator types on frequency-domain feature modeling, we applied Res-Blocks (He et al., 2016) and Swin Transformer blocks (Liu et al., 2021), both widely used for spatial-domain tasks, to

*Table 4.* Ablation on branch fusion strategies on GF2. Best in **bold**, second-best underlined.

| Fusion Strategy | PSNR↑ | SAM↓ | ERGAS↓ | FLOPs (G)↓ | Params (M)↓ |
|---|---|---|---|---|---|
| Average | 50.30 | 0.72 | 0.86 | **12.62** | **0.26** |
| Cross-Attention | 50.08 | 0.74 | 0.88 | 17.04 | 0.34 |
| Concat | **50.46** | **0.71** | **0.85** | 14.92 | 0.31 |
| $w_g$ Global | 50.40 | **0.71** | **0.85** | **12.62** | **0.26** |
| **Ours** | 50.30 | 0.72 | 0.86 | **12.62** | **0.26** |

process complex frequency-domain features. As shown in Table 5, despite the larger number of parameters and higher computational overhead of ResBlocks and Swin blocks, their performance in the frequency domain remains inferior to our Butterworth gating mechanism. These experimental results confirm that our Butterworth operator, designed based on classical signal processing theory, achieves more efficient and precise spectral modulation with lower parameter and computational costs.

*Table 5.* Comparison of different operators for frequency-domain feature modeling on the WV3 dataset.

| Method | PSNR↑ | ERGAS↓ | SAM↓ | Params (M)↓ | FLOPs (G)↓ |
|---|---|---|---|---|---|
| FFT + ResConv | 37.14 | 2.25 | 3.29 | 0.26 | 14.91 |
| FFT + Swin | 36.62 | 2.38 | 3.41 | 0.26 | 14.86 |
| **Ours (Full)** | **37.65** | **2.13** | **3.17** | **0.26** | **12.64** |

### 5.6. Ablation on Global Parameter Predictor Design

We compare four designs for the parameter predictor in the Global Filtering Path on GF2, as shown in Table 6. GAP+GMP introduces sensitivity to local noise, while

ResNet adds parameters without performance gains. Spatial-variant prediction (per-frequency) discards the physical prior of the Butterworth operator by breaking the assumption of globally coherent filter parameters. Our GAP+MLP achieves the highest PSNR of 50.30 dB, confirming that global average statistics provide the most stable frequency energy estimates for this low-dimensional regression task.

*Table 6.* Ablation on parameter predictor design for the Global Filtering Path (GF2).

| Predictor | PSNR↑ |
|---|---|
| GAP+GMP | 49.93 |
| ResNet | 49.97 |
| Spatial-variant (per-freq) | 50.05 |
| **GAP+MLP (Ours)** | **50.30** |

### 5.7. Anisotropic Butterworth vs. Anisotropic Gaussian Gating

We extend the conventional Gaussian filter to an anisotropic version and evaluate it under the same ABFNet architecture on the GF-2 dataset for a fair comparison. As shown in Table 7, the anisotropic Gaussian underperforms across all metrics, confirming that the specific mathematical form of the Butterworth filter is more effective for frequency-domain gating.

*Table 7.* Quantitative comparison between Anisotropic Gaussian and Butterworth gating on the GF-2 dataset.

| Mechanism | PSNR ↑ | ERGAS ↓ | SAM ↓ |
|---|---|---|---|
| Anisotropic Gaussian | 50.0020 | 0.7455 | 0.8867 |
| **Ours** | **50.3013** | **0.7201** | **0.8627** |

### 5.8. Generalization Analysis: CIFAR-100 Image Classification

*Table 8.* Performance comparison on CIFAR-100 under identical training settings. Best results in **bold**, second-best underlined.

| Method | Params (M)↓ | FLOPs (G)↓ | Acc (%) ↑ |
|---|---|---|---|
| ViT-Base (Dosovitskiy et al., 2021) | 5.36 | **0.35** | 69.00 |
| ConvNeXt-Tiny (Liu et al., 2022) | **5.17** | 0.75 | 79.44 |
| GFNet (Rao et al., 2021) | 6.26 | 0.80 | 80.30 |
| DINO-V3 (Siméoni et al., 2025) | 6.52 | 0.44 | 72.63 |
| VMamba-V2 (Liu et al., 2024) | 10.42 | 0.66 | 72.28 |
| UniRepLK (Ding et al., 2024) | 6.12 | 0.62 | 75.11 |
| MobileNet-V4 (Qin et al., 2024) | 6.26 | 0.76 | 81.25 |
| TransNext (Shi, 2024) | 6.59 | 1.01 | 80.79 |
| **Ours** | 5.54 | 0.70 | **81.31** |

To evaluate the generalization capability of our method beyond low-level reconstruction tasks, we employ the proposed BAM as a feature extraction backbone for the CIFAR-100 image classification benchmark. As shown in Table 8, with a comparable parameter scale, our model achieves a Top-1 accuracy of 81.31%, outperforming established spatial-domain backbones such as ConvNeXt-Tiny (Liu et al., 2022). Notably, our BAM also surpasses the frequency-domain competitor GFNet (Rao et al., 2021) by 1.01% in accuracy while maintaining lower computational overhead. This result validates that the anisotropic gating mechanism is not only adept at recovering fine-grained spatial details in pansharpening but also highly effective in capturing discriminative semantic features for high-level visual recognition tasks. For more operation details, please refer to Supplementary Material S6.

## 6. Conclusion

In this paper, we propose ABFNet, a novel frequency-aware framework that bridges the gap between classical signal processing and deep learning for pansharpening. By deconstructing the mathematical form of the Butterworth filter, we exhibit its profound structural correspondence with self-attention, reinterpreting it as an efficient, coordinate-based frequency gating mechanism. Furthermore, we introduce the BTI to enable global dependency modeling among tokens with $O(N \log N)$ complexity, replacing the expensive computations of standard Transformers. Extensive experiments on WV3, GF2, and QB datasets demonstrate that ABFNet significantly outperforms SOTA methods; meanwhile, its superior performance on CIFAR-100 classification further confirms its strong generalization potential across diverse vision tasks.

## Impact Statement

This paper presents a frequency-domain pansharpening framework with broad applicability in remote sensing and general computer vision. While enhanced Earth observation benefits civilian applications such as environmental monitoring, disaster relief, and urban planning, it carries an inherent dual-use risk: without proper regulatory oversight, high-fidelity satellite imagery could be misused for unauthorized surveillance or privacy infringement. Algorithmically, ABFNet's reliance on the smooth roll-off of the Butterworth filter may be suboptimal for extreme degradation scenarios requiring sharper frequency cutoffs. Future work will explore extending our parametric framework to other classical filter families, aiming to establish a more flexible, unified paradigm for frequency-domain feature modeling.

## Acknowledgements

This work was supported by the National Natural Science Foundation of China (No. 62376228), the Natural Science Foundation of Xinjiang Uyghur Autonomous Region, China (No. 2024D01A18 and No. 2024D01B07), the Sichuan Province Science and Technology Support Program, China

(No. 2025ZNSFSC1505), the Chengdu Science and Technology Program (No. 2025-YF12-00009-RC), and the Fundamental Research Funds for the Central Universities, Youth Faculty Development Program (No. JBK202511076).

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

# Supplementary Material

## S1. Theoretical Analysis

### S1. Theoretical Unification with Self-Attention

In this section, we establish a rigorous theoretical connection between our proposed Anisotropic Butterworth Module and the Self-Attention (SA) mechanism. We demonstrate that our method fundamentally operates as a Content-Adaptive Global Token Mixer unified under the framework of Dynamic Circulant Matrix Multiplication, incorporating a unique inductive bias towards directional smoothness.

### S1.1. UNIFIED FORMULATION: FROM ATTENTION TO CIRCULANT CONVOLUTION

Let $\mathbf{X} \in \mathbb{R}^{N \times C}$ be the input feature map, flattened from a spatial resolution of $H \times W$ into $N = H \cdot W$ tokens, where $C$ denotes the channel dimension. A generalized global token mixer—a mechanism allowing information exchange between any spatial positions—can be formulated as a linear transformation:

$$\mathbf{Y} = \mathbf{M}(\mathbf{X}) \cdot \mathbf{X} \tag{S1}$$

where $\mathbf{Y} \in \mathbb{R}^{N \times C}$ is the output feature, and $\mathbf{M}(\mathbf{X}) \in \mathbb{R}^{N \times N}$ is a data-dependent mixing matrix generated from the input $\mathbf{X}$.

**In Vision Transformers (ViT) [1]:** The mixing matrix $\mathbf{M}$ corresponds to the dense attention matrix $\mathcal{A}$, defined as:

$$\mathbf{M}_{SA} = \text{Softmax}\left(\frac{\mathbf{Q}\mathbf{K}^T}{\sqrt{d}}\right) \tag{S2}$$

where $\mathbf{Q}, \mathbf{K} \in \mathbb{R}^{N \times d}$ are Query and Key projections of $\mathbf{X}$. This matrix learns pairwise semantic similarities between tokens.

**In Our Method:** Based on the **Convolution Theorem** (which states that multiplication in the frequency domain equals convolution in the spatial domain [2]), our frequency domain modulation is equivalent to a spatial convolution with a global kernel $\mathcal{K}$. This operation can be strictly rewritten as a matrix multiplication where $\mathbf{M}$ takes the specific

form of a **Circulant Matrix** $\mathbf{C}_{\mathcal{K}}$:

$$\mathbf{Y} = \mathbf{C}_{\mathcal{K}}(\mathbf{X}) \cdot \mathbf{X} = \begin{bmatrix} \mathcal{K}_0 & \mathcal{K}_{N-1} & \dots & \mathcal{K}_1 \\ \mathcal{K}_1 & \mathcal{K}_0 & \dots & \mathcal{K}_2 \\ \vdots & \vdots & \ddots & \vdots \\ \mathcal{K}_{N-1} & \mathcal{K}_{N-2} & \dots & \mathcal{K}_0 \end{bmatrix} \begin{bmatrix} X_0 \\ X_1 \\ \vdots \\ X_{N-1} \end{bmatrix} \tag{S3}$$

Here, the kernel vector $[\mathcal{K}_0, \dots, \mathcal{K}_{N-1}]$ represents the spatial impulse response, dynamically generated via $\mathcal{K} = \mathcal{F}^{-1}(H(\mathbf{X}))$, where $\mathcal{F}^{-1}$ denotes the Inverse Discrete Fourier Transform (IDFT) and $H(\mathbf{X})$ is the learned frequency response. The Circulant Matrix is a special Toeplitz matrix where each row is a cyclic shift of the previous one, encoding the convolution operation in linear algebraic form [3].

### S1.2. MATHEMATICAL VERIFICATION OF THE GLOBAL RECEPTIVE FIELD

A critical theoretical question is whether the structured Circulant Matrix $\mathbf{C}_{\mathcal{K}}$ provides a true global receptive field—meaning every output pixel depends on every input pixel—comparable to the unconstrained attention matrix in ViT, or if it degenerates into a local operator (like a sparse matrix in standard CNNs). We prove the former by analyzing the analytic properties of the Butterworth kernel.

**1. Frequency Domain Definition**

Recall that our Anisotropic Butterworth filter $H(u, v)$ is defined as a **rational function** (a ratio of two polynomials) in the frequency domain:

$$H(u, v) = \frac{1}{1 + (\mathcal{D}(u, v; \theta)/D_0)^{2n}} \tag{S4}$$

where $u, v$ are spatial frequency coordinates, and $\mathcal{D}(u, v; \theta)$ is the oriented distance metric (e.g., Mahalanobis distance) rotated by angle $\theta$. $D_0$ is the cutoff frequency, and $n$ is the filter order. Mathematically, $H(u, v)$ is a strictly positive rational function defined over the entire frequency domain.

**2. Theoretical Derivation of Infinite Support**

To determine the spatial support (the region where the function is non-zero) of $\mathcal{K}$, we consider the Inverse Fourier Transform of $H(u, v)$. Since $H(u, v)$ is a rational function, its analytic structure in the complex plane is determined by its poles (points where the function value tends to infinity).

According to the Residue Theorem in complex analysis [4], the Inverse Fourier Transform of such a rational function is composed of the sum of the residues at these poles. For a Butterworth filter, the spatial impulse response $\mathcal{K}(x, y)$ generally takes the form of a sum of decaying exponentials and sinusoids:

$$\mathcal{K}(r) \approx \sum_{k=1}^{n} A_k e^{\lambda_k r} \qquad (S5)$$

where $r$ represents Euclidean spatial distance, $A_k$ are complex coefficients, and $\mathrm{Re}(\lambda_k) < 0$ ensures stability (decay).

### 3. Conclusion on Matrix Density
A fundamental property of the exponential function $e^{\lambda r}$ is that it **never vanishes** for any finite distance $r$ (it only asymptotically approaches zero). This implies:

$$|\mathcal{K}(x, y)| > 0, \quad \forall (x, y) \in \mathbb{R}^2 \qquad (S6)$$

In the discrete domain, this guarantees that the generated Circulant Matrix $\mathbf{C}_{\mathcal{K}}$ is a Dense Matrix (a matrix with no zero elements):

$$(\mathbf{C}_{\mathcal{K}})_{i,j} \neq 0, \quad \forall i, j \in \{1, \ldots, N\} \qquad (S7)$$

**Proof of Global Interaction:** Since every element in the mixing matrix is non-zero, the computation of any single output pixel $Y_i = \sum_j (\mathbf{C}_{\mathcal{K}})_{i,j} X_j$ physically aggregates information from **every** input pixel $X_j$ in the feature map. Thus, our module possesses a theoretically guaranteed global receptive field.

S1.3. Inductive Bias Analysis: Directional Selectivity and Shift-Invariance

While both SA and our method utilize dense matrices, the structure of $\mathbf{C}_{\mathcal{K}}$ introduces specific Inductive Biases—the set of assumptions a learning algorithm makes to predict outputs for unseen data [5]. We analyze these biases regarding Directional Preference and Shift-Invariance.

### 1. Explicit Directional Preference (Anisotropy)
In our formulation, the kernel $\mathcal{K}$ is not **isotropic** (invariant to rotation). It is the inverse transform of an **oriented** Butterworth filter $H(u, v; \theta)$. Geometrically, due to the rotation property of the Fourier Transform, if the frequency response $H$ is an ellipse oriented at angle $\theta$, the spatial kernel $\mathcal{K}$ will be an ellipse oriented at angle $\theta + \pi/2$ (orthogonal to the frequency orientation). This implies that the Circulant Matrix $\mathbf{C}_{\mathcal{K}}$ is structurally biased to aggregate information along a specific axis defined by $\theta$, while suppressing interference from orthogonal directions.

- **Comparison with SA:** While standard SA can learn to attend to specific directions implicitly, it starts as a "Tabula Rasa" (blank slate) and requires massive data

to discover this geometric prior. Our method explicitly injects this directional bias through the parameter $\theta$, forcing the matrix $\mathbf{C}_{\mathcal{K}}$ to facilitate anisotropic global mixing. This is particularly advantageous for Pansharpening, where features (edges, textures) possess strong directionality.

### 2. Shift-Invariance as a Structural Prior
The most profound difference lies in the matrix structure itself:

- **Self-Attention (Unconstrained):** The matrix $\mathbf{M}_{SA}$ is unconstrained, meaning the attention weight $A_{i,j}$ (between pixel $i$ and $j$) can be completely different from $A_{i+1,j+1}$ (between their neighbors). This lack of constraint allows SA to model arbitrary relationships but sacrifices Shift-Invariance—a fundamental property of visual data where a feature (e.g., a corner) should be processed similarly regardless of its absolute position in the image [6].

- **Ours (Toeplitz/Circulant):** The definition of the Circulant Matrix enforces that $\mathbf{M}_{i,j} = \mathbf{M}_{i+1,j+1} = \mathcal{K}_{i-j}$. This imposes a strict Shift-Invariance Prior.

- **Inductive Bias:** This structure forces the model to learn a rule based on relative position ($\Delta = i - j$) rather than absolute position indices.

- **Benefit:** This acts as a strong regularizer. It ensures that the "global mixing logic" learned by the network is spatially consistent across the entire image. For restoration tasks, this consistency is vital to avoid generating location-dependent artifacts (such as blocking effects or seams), ensuring that the restoration quality is uniform across the image plane.

S1.4. Spatial Duality

A key theoretical concern is whether applying spatial operators (convolution kernel $K[m]$ or learned attention weights) directly to the frequency-domain representation is equivalent to our Butterworth gating. Via the convolution theorem, the two paradigms diverge fundamentally in their spatial-domain semantics. For a spatial operator (SpecOps), frequency-domain mixing yields:

$$y_{\mathrm{spec}}[n] = \mathcal{F}^{-1}\mathrm{Mix}(X[k]) = x[n] \cdot M[n], \qquad (S8)$$

where $M[n] = \sum_m K_{\mathrm{dyn}} e^{j \frac{2\pi}{N} mn}$ is the data-dependent modulation term. Taking the real part reduces this to $x[n] \cdot A(n) \cos \phi(n)$, where the residual $\cos \phi(n)$ term induces spatial gain oscillation, violating translation invariance and collapsing to a coordinate-dependent point-wise scalar that cannot model neighborhood context. By contrast,

Butterworth frequency-domain multiplication yields a true spatial convolution:

$$y_{\text{ours}}[n] = \mathcal{F}^{-1}\big(X[k] \cdot H[k]\big) = (x \circledast h)[n], \qquad \text{(S9)}$$

where $h$ is the Butterworth impulse response and $(x \circledast h)[n] = \sum_m x[m]\,h[n-m]$. This ensures the Linear Time-Invariant (LTI) property and structural integrity of the spatial aggregation, providing the neighborhood context essential for high-frequency detail recovery in pansharpening.

## S2. Implementation Details

The proposed ABFNet was implemented using the PyTorch deep learning framework and trained/tested on a single NVIDIA RTX 5060 Ti GPU. The network was trained for 350 epochs with a batch size of 16. Parameter optimization was performed using the Adam optimizer with an initial learning rate of $1 \times 10^{-3}$. To ensure training stability and convergence, we employed a Multi-Step learning rate decay strategy; specifically, the learning rate was decayed by a factor of 0.5 at epochs [100, 200, 300, 320]. In the data pre-processing stage, input images were randomly cropped into $64 \times 64$ patches, and data augmentation techniques such as random flipping and rotation were applied to prevent overfitting.

## S3. Computational Complexity Analysis

To provide a rigorous comparison of computational efficiency, we analyze the complexity of the standard Self-Attention mechanism and our proposed Butterworth Attention Module (BAM). Let $N$ denote the total number of tokens (or patches) in the feature map, and $d$ represent the feature dimension (embedding size).

### S3.1. Complexity of Standard Self-Attention

The standard Self-Attention mechanism involves three primary stages: linear projections, affinity matrix computation, and value aggregation. The detailed computational breakdown is as follows: First, the input feature $X \in \mathbb{R}^{N \times d}$ is projected into Query ($Q$), Key ($K$), and Value ($V$) matrices using learnable weights $W_Q, W_K, W_V \in \mathbb{R}^{d \times d}$. This operation requires:

$$T_{\text{proj}} = 3 \times (N \times d \times d) = 3Nd^2. \qquad \text{(S10)}$$

Second, the attention score matrix is computed via the inner product of $Q$ and $K^T$, resulting in an $N \times N$ matrix. The complexity of this matrix multiplication is:

$$T_{\text{score}} = N \times d \times N = N^2 d. \qquad \text{(S11)}$$

Third, the Softmax-normalized attention matrix is multiplied

by $V$ to produce the final output, which similarly involves an $N \times N$ matrix multiplied by an $N \times d$ matrix:

$$T_{\text{agg}} = N \times N \times d = N^2 d. \qquad \text{(S12)}$$

Summing these components, the total complexity of Self-Attention is:

$$T_{\text{Total\_SA}} = 3Nd^2 + 2N^2 d \approx \mathcal{O}(N^2 d). \qquad \text{(S13)}$$

Given that $d$ is typically a constant hyperparameter, the complexity scales quadratically as $\mathcal{O}(N^2)$ relative to the number of tokens.

### S3.2. Complexity of BAM

In contrast, the BAM facilitates global interaction through frequency-domain modulation. Let the parameter manifold be arranged as a 2D map $\mathcal{M} \in \mathbb{R}^{\sqrt{N} \times \sqrt{N} \times C_p}$, where $C_p$ is the number of filter parameters. The computational steps are as follows:

First, the 2D FFT is applied to the parameter map to project it into the spectral domain. The complexity of a 2D FFT for $N$ points is:

$$T_{\text{FFT}} = \mathcal{O}(C_p \cdot N \log N). \qquad \text{(S14)}$$

Second, the interaction is performed via element-wise gating with the learnable Butterworth mask $H \in \mathbb{C}^{\sqrt{N} \times \sqrt{N}}$. Since this is a Hadamard product, the complexity is linear:

$$T_{\text{gating}} = \mathcal{O}(C_p \cdot N). \qquad \text{(S15)}$$

Third, the modulated spectrum is transformed back to the spatial domain using the Inverse 2D FFT (IFFT), which has the same complexity as the forward transform:

$$T_{\text{IFFT}} = \mathcal{O}(C_p \cdot N \log N). \qquad \text{(S16)}$$

The total complexity of the BAM is:

$$T_{\text{Total\_BAM}} = \mathcal{O}(2 \cdot C_p N \log N + C_p N) \approx \mathcal{O}(N \log N). \qquad \text{(S17)}$$

### S3.3. Comparison and Conclusion

Comparing the two paradigms, while Self-Attention relies on exhaustive pairwise computations resulting in $\mathcal{O}(N^2)$ complexity, BAM leverages the global receptive field of the FFT. By replacing matrix-based affinity with coordinate-aware spectral gating, BAM achieves comprehensive patch interaction while reducing the computational burden to $\mathcal{O}(N \log N)$, making it significantly more scalable for high-resolution remote sensing imagery.

## S4. Physical Constraints and Parameter Range Analysis

The learnable parameters in ABFNet are not directly optimized in their raw form. Instead, they undergo specific non-linear transformations to ensure they remain within physically meaningful bounds while maintaining differentiability. This mapping mechanism is crucial for the stability of frequency-domain learning.

### S4.1. Frequency Cutoff Constraints

The radial cutoff frequencies $D_{0u}$ and $D_{0v}$ determine the passband of the elliptical filter. In our implementation, these are constrained to the range $[0.01, 0.8]$ using the mapping $0.01 + 0.79 \cdot \sigma(\cdot)$. The lower bound of $0.01$ prevents the filter from zeroing out the DC component (the $(0, 0)$ frequency), which contains the mean intensity of the image and is vital for structural integrity. The upper bound of $0.8$ ensures that the filter does not attempt to modulate frequencies beyond the Nyquist limit, thereby avoiding aliasing artifacts.

### S4.2. Directional Orientation and Rotation

The rotation angle $\theta$ is mapped to the interval $[-\pi/2, \pi/2]$ via $(\sigma(\theta) - 0.5) \cdot \pi$. This range is sufficient to cover all possible orientations of an elliptical filter in a 2D spectral plane due to the symmetry of the FFT. By allowing $\theta$ to be learnable, the network can align the major and minor axes of the filter with the dominant texture orientations (e.g., urban grids or natural edges) in remote sensing imagery.

### S4.3. Steepness and Filter Order

The filter order $n$ controls the transition band's slope. We constrain $n$ within $[0.5, 4.0]$ using $0.5 + 3.5 \cdot \sigma(n)$. At $n = 0.5$, the Butterworth filter exhibits a very soft, Gaussian-like decay, which is beneficial for smooth spectral blending. As $n$ approaches $4.0$, the response becomes increasingly steep, acting as a near-ideal filter that can sharply isolate specific high-frequency details or suppress noise. This flexibility allows each channel to autonomously determine its selectivity level.

### S4.4. Dual Mode Gating Logic

The gate parameter $g$ undergoes a standard Sigmoid transformation to the range $[0, 1]$. It serves as a soft toggle for the dual-mode mixing mechanism:

$$H_{\text{filter}} = g \cdot (1.0 - \text{base}) + (1.0 - g) \cdot \text{base}. \qquad \text{(S18)}$$

When $g \approx 0$, the module operates in a Low-Pass Filtering mode, emphasizing structural coherence and smoothing. When $g \approx 1$, the module switches to a High-Pass Filtering mode, focusing on extracting fine textures and edges from the PAN image. The learnable nature of $g$ enables the model to perform dynamic frequency selection based on the local or global context of the input features.

### S4.5. Coordinate System and Grid Generation

To ensure the filter is correctly centered in the frequency domain, we generate a normalized coordinate grid using `rfftfreq` and `fftfreq`. These coordinates are then rotated using a standard 2D rotation matrix derived from $\theta$. The distance metric $\mathcal{R}$ is calculated as an elliptical ratio, allowing for anisotropic modulation. The inclusion of a small epsilon ($\epsilon = 1e - 8$) during the ratio and log calculations prevents numerical instability and ensures robust gradient propagation during backpropagation.

## S5. Distribution of Learnable Parameter $n$

To investigate the behavior of the learnable anisotropic Butterworth filters, we visualize the distribution of the filter order $n$ across the cascading layers of ABFNet. As shown in Fig. S1, the network does not converge to a uniform value; instead, it learns distinct frequency-selectivity profiles for different feature hierarchies.

## S6. Detailed Architecture for CIFAR-100 Classification

To bridge the gap between low-level pixel reconstruction and high-level semantic recognition, we adapt ABFNet into a pyramidal hierarchical backbone. While the pansharpening task necessitates the preservation of full-resolution spatial structures, the classification paradigm requires the model to progressively aggregate local information into abstract, category-discriminative representations. This is achieved by systematically downsampling the spatial resolution while expanding the channel dimensionality to increase semantic density.

### S6.1. Macro-Architecture and Hierarchical Design

The classification variant, denoted as ButterFlowNet-CIFAR, consists of a shallow patch embedding layer followed by three hierarchical stages. Each stage is composed of multiple Butterworth Attention Module. Between stages, a downsampling module (comprising a $3 \times 3$ convolution with stride 2 and Layer Normalization) is employed to reduce the spatial resolution and double the feature channels. The detailed configuration, optimized for the $32 \times 32$ input scale of the CIFAR-100 dataset, is presented in Table S1. This architecture ensures that the global Butterworth filters in the deeper stages can operate on high-level feature maps, effectively capturing the global spectral envelope of semantic objects.

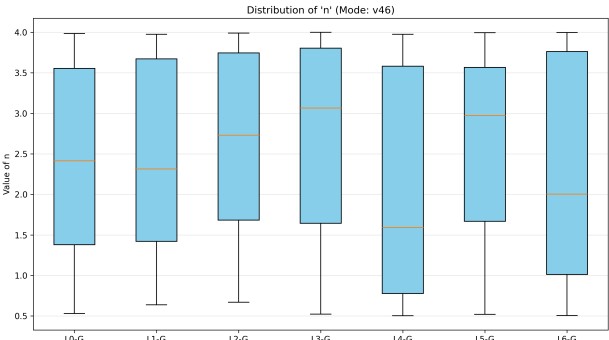

*(a)* Layer-wise distribution of $n$

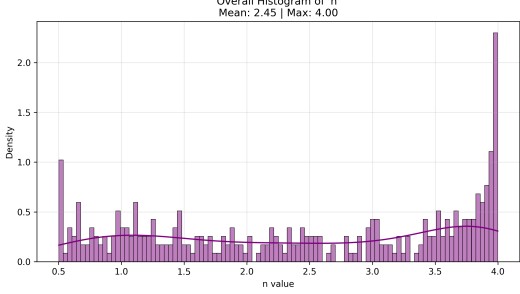

*(b)* Overall density of $n$

*Figure S1.* Statistical analysis of the learnable filter order $n$. (a) The boxplot illustrates how the network adjusts the steepness of the Butterworth response across different levels (L0-G to L6-G), with the median value (orange line) shifting to meet the requirements of varied spatial-spectral scales. (b) The overall histogram reveals a mean value of $2.45$ and a concentration of higher values near $4.00$, indicating that the model frequently utilizes steep transition bands to isolate high-frequency textures.

## S6.2. Implementation of Adaptive Frequency Modulation

Unlike standard Vision Transformers that rely on fixed positional encodings, the classification variant of ABFNet utilizes the coordinate-aware nature of the Butterworth filter to provide an inherent frequency-domain inductive bias.

**Global Branch:** In the deeper stages (e.g., Stage 3), the global anisotropic filter modulates the $8 \times 8$ spectral map. At this stage, the learnable cutoff frequencies $(D_{0u}, D_{0v})$ act as semantic gatekeepers, selectively passing frequency components that correspond to the holistic shape and primary orientation of the object categories.

**Local Branch:** The $8 \times 8$ window partitioning remains constant across all stages. As the resolution decreases, each window covers a larger receptive field relative to the original image. The Butterworth Token Interaction (BTI) module facilitates $O(N \log N)$ interaction between these windows, enabling the model to reconcile local texture details with the broader semantic context without the quadratic complexity of self-attention.

*Table S1.* Macro-architecture of ButterFlowNet for CIFAR-100.

| Stage | Layer Type | Output Size | Repeat ($L$) |
|---|---|---|---|
| Input | Image | $32 \times 32 \times 3$ | – |
| Stage 1 | Patch Embed | $32 \times 32 \times 64$ | 1 |
| | BAM | $32 \times 32 \times 64$ | 3 |
| Stage 2 | Downsample | $16 \times 16 \times 128$ | 1 |
| | BAM | $16 \times 16 \times 128$ | 9 |
| Stage 3 | Downsample | $8 \times 8 \times 256$ | 1 |
| | BAM | $8 \times 8 \times 256$ | 3 |
| Classifier | Global AvgPool | $1 \times 1 \times 256$ | 1 |
| | FC Head | 100 | 1 |

### S6.3. Block-Level Specification

Each BAM integrates a dual-path frequency modulation strategy. The Global Branch performs a FFT on the entire feature map, applying a single set of learnable anisotropic Butterworth parameters to capture the holistic spectral envelope of the image. Simultaneously, the Local Branch partitions the feature map into $8 \times 8$ non-overlapping windows.

### S6.4. Complexity and Classifier Head

The final output of Stage 3 is processed by a Global Average Pooling (GAP) layer, followed by Layer Normalization and a Linear Head to project the 256-dimensional feature vector to the 100-class probability distribution. Despite the deep 15-block hierarchy, the model maintains a low computational footprint of approximately 0.70G FLOPs, primarily due to the linear complexity of the Butterworth gating mechanism compared to the quadratic cost of standard self-attention.

## S7. Statistical Analysis of the Gating Parameter $g$

To investigate the actual behavior of the learned gating parameter $g$, we collect the channel-wise values of $g$ from the Global and Local branches across three representative ABFNet blocks (Block 1, 4, and 7) on the GF2 dataset. As reported in Table S2, the statistics reveal two key findings. 1) Adaptive passband allocation. Across all blocks and branches, the mean value of $g$ consistently exceeds 0.5 (ranging from 0.601 to 0.919), which indicates that the network predominantly operates in a high-pass-biased mode. The module learns to adaptively allocate frequency passbands according to the specific spatial-spectral demands of each layer. 2) Dynamic per-channel frequency selection. The standard deviations of $g$ are consistently large, with values spanning the full range $[0, 1]$ across channels. This high

channel-wise variance confirms that different feature channels autonomously select distinct frequency modes. This behavior is precisely the motivation for the per-channel design of $g$ over a single global scalar, complementing the branch fusion ablation in the main paper.

*Table S2.* Channel-wise statistics of the learned gating parameter $g$ across three representative blocks and both branches.

| Block | Branch | Mean | Std | Max | Min |
|---|---|---|---|---|---|
| Block 1 | Global | 0.9189 | 0.1976 | 0.9983 | 0.0066 |
| | Local | 0.7583 | 0.3155 | 1.0000 | 0.0000 |
| Block 4 | Global | 0.8606 | 0.2700 | 0.9992 | 0.0029 |
| | Local | 0.8022 | 0.2794 | 1.0000 | 0.0000 |
| Block 7 | Global | 0.8762 | 0.2266 | 0.9970 | 0.0086 |
| | Local | 0.6007 | 0.2653 | 1.0000 | 0.0001 |

