# OpenReview forum: "Butterworth as Attention: Anisotropic Spectral Gating for Pansharpening"
_ICML.cc/2026/Conference — ICML 2026 regular_

### Official Review · Reviewer_Bp1N · 2026-03-03

**Soundness:** 3
**Presentation:** 3
**Significance:** 2
**Originality:** 2
**Overall Recommendation:** 4
**Confidence:** 5

**Summary:**

This paper proposes a frequency-domain gating framework for efficient pansharpening. It parameterizes the Butterworth filter and further introduces anisotropic parameters and a BTI operation to enable effective feature fusion and interaction.

**Compliance With Llm Reviewing Policy:**

Affirmed.

**Final Justification:**

The authors have addressed all my concerns in the rebuttal, and I would like to upgrade my rating.

**Key Questions For Authors:**

See Weakness.

**Limitations:**

It would be helpful to include an analysis of why the method underperforms compared to other baselines on the QB full-resolution setting, as this may help identify its limitations and inspire future improvements.

**Strengths And Weaknesses:**

Strengths:
1. The paper is grounded in native frequency-domain filtering. It learns the filter parameters with a neural network and further extends the original formulation by introducing anisotropic parameters, which enhances its capability to fuse information in the frequency domain and leads to strong results across multiple datasets.
2. The experimental evaluation is fairly comprehensive. The authors test on several datasets and also provide comparisons against Gaussian filtering as well as common spatial-domain operations transplanted into the frequency domain.

Weaknesses：
1. It is unclear how the spatial-domain kernel in Fig. 4 is obtained.
2. To better substantiate the efficiency claim, the paper should report inference latency and GPU memory usage.
3. The method performs strongly on reduced-resolution settings, but is not as competitive on QB full-resolution. This raises the question of whether the approach has limitations on high-resolution imagery.
4. In my view, the main novelty lies in the anisotropic design and the BTI module, however, the ablation results suggest that the performance gains brought by these two components are relatively modest.

---

> ### Author Rebuttal · Authors · 2026-03-30
>
> **1. Regarding the spatial-domain kernel in Fig. 4**, we added a step-by-step explanation. We retrieve anisotropic parameters to reconstruct the frequency-domain mask $M(u,v)$ per channel. To obtain the spatial kernel, we apply `ifftshift` to reorder the quadrants, perform a 2D IFFT, then shift the zero frequency back to the centre with `fftshift` and take the real part. Because the Butterworth kernel concentrates its energy near the origin, we crop the central $48 \times 48$ region for visualization. The red arrow in the figure indicates the dominant texture direction of the input feature map, which is estimated as follows: Sobel operators compute the horizontal and vertical gradients $G_x, G_y$; for each pixel we calculate the gradient orientation $\arctan2(G_y, G_x)$ and map it to the range $[0^\circ, 180^\circ)$ (texture direction is modulo $180^\circ$); we then build a histogram with 36 bins (each bin covering $5^\circ$) and take the centre angle of the peak bin as the dominant texture direction. The cyan dashed line shows the major axis of the frequency passband, and the green arrow shows the long axis of the spatial kernel. Theoretically the spatial kernel should be elongated along the texture direction, so the green arrow aligns with the red arrow; the alignment observed in Fig. 4 confirms that the model learns a correct directional prior.
>
> **2. Regarding inference latency and GPU memory usage**, we added the following measurements  (tested on a single RTX 5060 Ti, batch size 1, FP32; MS: $64 \times 64$, PAN: $256 \times 256$).
>
> |Method|Inference Latency (ms) $\downarrow$|Peak GPU Memory (MB) $\downarrow$|
> |:---|:---:|:---:|
> |SFIIN|9.29|76.03|
> |TDNet|6.90|84.20|
> |BIMPan|192.75|342.23|
> |DISPnet|29.70|101.90|
> |DCINN|10.27|56.15|
> |WFAnet|OOM|OOM|
> |ARConv|92.56|671.58|
> |BNNPan|31.60|159.18|
> |Ours|38.19|297.89|
>
> Local CNNs (SFIIN, TDNet) achieve lower latency due to optimized $3 \times 3$ convolution kernels but lack long-range dependency modeling. Among methods that provide global  modeling, our model achieves a superior trade-off. WFAnet (Transformer) runs out of memory due to complexity. ARConv (dynamic convolution) consumes double the latency and peak memory of ours. Importantly, our asymptotic complexity $\mathcal{O}(N \log N)$ scales much better than self-attention models ($\mathcal{O}(N^2)$).
>
> **3. Regarding the concern about limitations on high-resolution imagery and the QB full-resolution performance:** Our model is trained on $64 \times 64$ PAN patches but tested on $256 \times 256$ PAN images for QB and GF2. Under this setting, ABFNet achieves significant advantages over existing methods (Table 1), demonstrating strong generalization to higher resolutions.
>
> **Regarding the marginal HQNR gap on full-resolution QB:** This stems from the inconsistency between Wald protocol's simulated degradation and real-world satellite degradation. For instance, ARConv achieves higher full-resolution (FR) HQNR but substantially lower reduced-resolution (RR) PSNR than ours. This metric inversion indicates no single metric fully characterizes performance. Pansharpening requires balancing RR metrics, FR metrics, and visual quality. Holistically, ABFNet achieves state-of-the-art RR results, remains competitive on FR metrics, and shows superior visual quality (Figs. 5, 6). The HQNR gap does not imply a high-resolution limitation.
>
> To demonstrate practical potential, we fine-tuned ABFNet using 48 FR samples (1 epoch, $L_1$ + HQNR loss). HQNR improved significantly while RR metrics remained unchanged (below). This confirms ABFNet effectively adapts to real data distributions with minimal supervision.
>
> |Method|PSNR|SAM|ERGAS|HQNR|$D_\lambda$|$D_s$|
> |:---|:---:|:---:|:---:|:---:|:---:|:---:|
> |Original|38.83|3.41|4.06|0.891|0.050|0.063|
> |Finetuned|38.86|3.39|4.05|0.933|0.032|0.037|
>
>
> **4. Regarding the modest gains of the anisotropic design and BTI :**
>
> Our primary contribution lies in reinterpreting the classical Butterworth filter as a learnable operator within a dual-branch architecture, not limited to these two components.
>
> As shown in Table 2, even without BTI or anisotropy, our core dual-branch framework already achieves state-of-the-art performance across all datasets. This proves the foundational effectiveness of our frequency-domain modeling.
>
> The BTI and anisotropic modules serve as refinements on this foundation.Their improvements—such as a 0.3 dB gain on GF2 from anisotropy with negligible parameter overhead—validate the importance of cross-patch interaction and directional perception. While incremental, they provide stable, low-cost enhancements to an already leading framework. We have revised the manuscript to clarify this contribution hierarchy.

---

> > ### Author Rebuttal · Reviewer_Bp1N · 2026-04-03
> >
> > The authors have addressed all my concerns in the rebuttal, and I would like to upgrade my rating.

---

> > > ### Author Response · Authors · 2026-04-04
> > >
> > > We sincerely appreciate the time and effort you have dedicated to reviewing our manuscript, as well as your positive recognition of our rebuttal.

---

### Official Review · Reviewer_mvvy · 2026-03-09

**Soundness:** 2
**Presentation:** 2
**Significance:** 3
**Originality:** 3
**Overall Recommendation:** 4
**Confidence:** 5

**Summary:**

This paper first identifies the core limitation in current pansharpening procedures. Specifically, existing approaches typically embed spatial domain operator modules directly into the network architecture, which lacks a structured investigation of spatial frequency domain characteristics. Motivated by this observation, we reconstruct an anisotropic Butterworth low pass filter that functions as a novel gated attention mechanism. We subsequently incorporate this module into a dual branch frequency domain framework to construct the Anisotropic Butterworth Fusion Network, referred to as ABFNet. Consequently, the proposed architecture performs pansharpening with linear computational complexity, which ultimately achieves state of the art precision across multiple datasets.

**Compliance With Llm Reviewing Policy:**

Affirmed.

**Final Justification:**

The additional experiments have addressed my concerns. Thus I decide to raise the score.

**Key Questions For Authors:**

(1)	I acknowledge that the proposed reconstruction of the Butterworth low pass filter possesses sufficient originality. However, interpreting this reconstruction as an attention mechanism remains unconvincing because the method does not emphasize token interaction or correlation modeling. I am willing to reconsider my score if the authors can refine and optimize the narrative of this specific section.

(2)	Within the Global Filter Path, the parameters $D_{u0}$, $D_{v0}$, $\theta$, $n$, and $g$ are crucial to the final performance. One might question whether generating and adjusting these variables solely through a multilayer perceptron is overly simplistic. I would be open to reevaluating my assessment if the authors could either enhance this architectural design or provide a compelling justification for relying exclusively on a multilayer perceptron.

(3)	Could the authors clarify the underlying reasoning behind the assertion in the abstract, which states that directly transplanting spatial operators in conventional methods neglects the fundamental structure of the spectrum along with its strict spatial correspondence? Furthermore, since the current manuscript lacks concrete examples or mathematical proofs to logically substantiate this claim, could you provide the specific theoretical derivations or empirical evidence required to support it?

**Limitations:**

This paper does not discuss the limitations of the current work nor potential future research directions. As noted in the weaknesses section, could future work attempt to transform and reconstruct the transfer functions of other filters, such as Chebyshev or elliptic filters, using the $x=e^{\ln x}$ identity? This could potentially establish a unified structural modeling paradigm for frequency-domain characteristics across the classical filter family, specifically tailored for pansharpening.
Furthermore, to handle the high-frequency information required for pansharpening, your paper employs a simple gating weight $g$ to perform linear interpolation between the low-pass mask $H(u,v)$ and the high-pass mask. This approach lacks mathematical persuasiveness. Would it be feasible to reconstruct this mechanism starting directly from the classical bandpass or highpass filter formulas?

**Strengths And Weaknesses:**

Strengths:

The inference process achieves a complete mathematical logical closure. Specifically, the study transforms the transfer function of the Butterworth low pass filter into an exponential form by utilizing the identity $x=e^{\ln x}$ and subsequently incorporates a rotation matrix for coordinate projection. This adaptation enables the framework to effectively capture directional high frequency textures that are inherent in remote sensing imagery. Such a rigorous and detailed mathematical derivation ensures that the entire analytical process maintains strict logical consistency.


Weaknesses:

（1）Regarding the formatting issues, I noticed that the initial word in Section 3.1 Butterworth Filter Revisited is misspelled as "he" instead of "The". Furthermore, the variables $D_0$ and $n$ are inconsistently italicized throughout the manuscript. These represent fundamental typographical errors that require careful proofreading.

（2）Although the mathematical derivation is logically rigorous, interpreting the dot product between $D_0$ and $n$ as an attention token interaction remains somewhat unconvincing. The product of these two variables fails to effectively demonstrate correlation modeling within the frequency domain. This operation functions more as a logical analogy that aims to illustrate frequency sensitivity and selective attention. Therefore, it would be more appropriate to interpret this mechanism as a dynamic filter driven by physical inspiration.

（3）The authors assert that the direct transplantation of spatial operators in conventional methods neglects the fundamental structure of the spectrum along with its strict spatial correspondence. However, the manuscript lacks the necessary theoretical proofs to substantiate this claim.

（4）Contemporary approaches do not uniformly resort to the straightforward transplantation of spatial operators. Specific methodologies perform elementwise multiplication by applying a learnable global filter that strictly aligns with the spectral resolution immediately after the two dimensional fast Fourier transform. Other strategies employ a polar coordinate attention mechanism that maps the amplitude of the scattering vector. Consequently, the current literature review appears to lack sufficient depth and comprehensiveness.

（5）Throughout the manuscript, there are numerous instances where a single isolated word occupies an entire line. Such a typesetting anomaly significantly deviates from rigorous academic formatting standards. Consequently, this issue severely impairs the overall readability and structural coherence of the paper.

---

> ### Author Rebuttal · Authors · 2026-03-31
>
> **1. Re: formatting issues:**
> We have corrected the typo in Sec. 3.1, italicized mathematical variables, and adjusted paragraph structures to eliminate isolated words. Thanks for pointing these out.
>
> **2. Re: interpreting the Butterworth filter as attention and token interaction:**
> The global branch acts as a dynamic filter, while the local branch performs **Parameter-level Interaction**, not just simple dynamic filtering.
> 1) For each patch, we predict initial filtering parameters to form a parameter token grid $T \in \mathbb{R}^{Nh \times Nw \times C}$.
> 2) We apply a second Butterworth operator in the frequency domain to $T$. This serves as a **global mixer**, allowing the filtering logic of one patch to be informed by all others.
> 3) The resulting refined parameters $T'$ encapsulate global structural context.
>
> Like the token mixing in Transformers, our module achieves **all-to-all interaction** among patches. The distinction lies in the inductive bias: instead of unconstrained $QK^T$ weights, we enforce a physical Butterworth prior. This ensures that parameter correlations follow natural frequency distributions, significantly enhancing stability in pansharpening (Table 2).
>
> **3. Re: generating parameters solely through an MLP:**
> The Global Filter Path captures global characteristics, not local features. An MLP on global pooled features acts as a hypernetwork for this low-dimensional regression. Adding complexity (CNNs/attention) introduces redundancy and instability, as confirmed on GF2:
>
> |Predictor|PSNR|
> |:---|:---:|
> |GAP+GMP|49.93|
> |ResNet|49.97|
> |Spatial-variant (per-freq)|50.05|
> |GAP+MLP(ours)|50.30|
>
> Global Average Pooling (GAP) provides stable energy estimates aligned with the Butterworth operator. Global Max Pooling (GMP) introduces sensitivity to local noise. Spatial-variant loses the physical prior. ResNet adds complexity without gains. Our GAP+MLP is optimal.
>
> **4. Re: transplanting spatial operators into the frequency domain:**
> We clarify the theoretical gap between SpecOps (Conv. $K[m]$ / Attention) and our Butterworth filtering ($H[k]$) via spatial duality.
> **1) Baseline (SpecOps):** Freq-mixing ($X * K$) yields a spatial point-wise scaler:
> $$y_{spec}[n] = \mathcal{F}^{-1}\{ \text{Mix}(X[k]) \} = x[n] \cdot \underbrace{\sum K_{dyn} e^{j\frac{2\pi}{N}mn}}_{M[n]} \xrightarrow{\text{Real}} x[n] \cdot A(n)\cos\phi(n)$$
> where $\cos\phi(n)$ induces **spatial gain oscillation**. This violates translation invariance and lacks neighborhood context.
>
> **2) Ours (Butterworth):** Freq-multiplication ($X \cdot H$) yields a spatial convolution:
> $$y_{ours}[n] = \mathcal{F}^{-1}\{ X[k] \cdot H[k] \} = \underbrace{\sum_{m} x[m] h[n-m]}_{(x \circledast h)[n]} \xrightarrow{\text{LTI}} \text{Spatial Aggregation}$$
> where $h$ is the Butterworth impulse response. This ensures **LTI (Linear Time-Invariant)** property and structural integrity.
>
> **Conclusion:** SpecOps collapse to **coordinate-dependent masks**, failing to model the neighborhood dependencies $(x \circledast h)$ essential for pansharpening. Our approach satisfies this requirement within a unified physical framework.
>
> **5. Re: limitations, the gating mechanism, and future directions:**
> We extend the framework to other filters via a unified parametric formulation:
> $$H_{UPF}(D) = (1-g) \cdot \frac{1}{1+\epsilon^2 \cdot (\sum_{i \in \{x,y\}} |\text{rot}(k_i)D_{l,i}|^p)^{2n/p}} + g \cdot \frac{1}{1+\epsilon^2 \cdot (\sum_{i \in \{x,y\}} |\text{rot}(k_i)D_{h,i}|^p)^{2n/p}}$$
> The gating weight $g$ enables bandpass behavior. This subsumes Butterworth ($p=2, \epsilon=1$), Chebyshev (via $\epsilon$), and elliptical shapes (via $p$). Testing on GF2 below show other filters show marginal differences (<0.2 dB). While Table metrics are competitive, the complex gradients of other filters often trigger **NaN** during training.
> |Method|PSNR|
> |:---|:---:|
> |SuperElliptic|50.21|
> |Chebyshev|50.28|
> |Unified|50.14|
> |Ours|50.30|
>
> **6. Re: Literature Review Depth**
> We restructure Sec 2.2 into four paradigms: (a) Spatial transplantation; (b) Unconstrained global filters (e.g., GFNet); (c) Coordinate-transformed attention (e.g., Polar-based); (d) Our physically-inspired parametric modeling.
>
> **(1) vs. Unconstrained Filters:** While GFNet uses element-wise multiplication, its heuristic weights are tied to a fixed grid, causing discretization mismatch across resolutions. Our BW predicts continuous, **resolution-independent** parameters ($D_0, n$), ensuring stable reconstruction across scales.
> | Method | WV3 | QB | GF |
> | :--- | :---: | :---: | :---: |
> | GFnet | 37.17 | 38.04 | 48.92 |
> | Ours | **37.65** | **38.83** | **50.30** |
>
> **(2) vs. Polar Attention:** Polar methods require coordinate resampling, leading to spectral leakage and geometric distortion. We retain **Cartesian consistency** and model anisotropy via parametric $(\theta, D_u, D_v)$. Per Point 4, BW ensures **structured spatial convolution**, whereas polar attention collapses to unstable point-wise scaling.

---

> > ### Author Rebuttal · Reviewer_mvvy · 2026-04-03
> >
> > I appreciate the authors’ clarifications. The additional experiments have addressed my concerns. Thus I decide to raise the score.

---

> > > ### Author Response · Authors · 2026-04-04
> > >
> > > We sincerely appreciate the time and effort you have dedicated to reviewing our manuscript, as well as your positive recognition of our rebuttal.

---

### Official Review · Reviewer_Eu9W · 2026-03-13

**Soundness:** 3
**Presentation:** 3
**Significance:** 3
**Originality:** 3
**Overall Recommendation:** 4
**Confidence:** 4

**Summary:**

This paper proposes ABFNet, a pansharpening network whose core contribution is a learnable, anisotropic Butterworth filter used as a drop-in replacement for self-attention in a Transformer-like architecture. The central theoretical insight is that the standard isotropic Butterworth filter can be algebraically reformulated as a Sigmoid-gated frequency-domain attention mechanism, with the cutoff frequency D₀ playing the role of a Query, the filter order 2n playing the role of a Key, and 2n·ln D(u,v) serving as a dynamic positional encoding. The paper then generalizes this to an anisotropic elliptical form (via coordinate rotation and per-axis cutoff frequencies, Du₀ and Dv₀), thereby enabling direction-aware spectral gating. This anisotropic filter is instantiated in a dual-branch Butterworth Attention Module (BAM): a global branch applies one filter to the full feature map, and a local branch uses a two-stage Butterworth Token Interaction (BTI) mechanism to generate per-patch filter parameters through frequency-domain communication between patch tokens. ABFNet achieves SOTA PSNR on the WV3, QB, and GF2 pansharpening benchmarks with a remarkably small parameter count (0.26M), and its core gating module also achieves competitive CIFAR-100 accuracy.

**Compliance With Llm Reviewing Policy:**

Affirmed.

**Final Justification:**

I will maintain my initial score.

**Key Questions For Authors:**

1. In the Butterworth-as-Attention analogy, D₀ and n are generated from global statistics (GAP + MLP). How do these compare to per-sample, per-frequency, or per-channel parameter generation? Have you explored making D₀ or n vary per-channel or per-spatial-location, and if so, what is the effect?

2. ABFNet's full-resolution HQNR (0.891) is lower than ARConv (0.915) despite superior reduced-resolution PSNR. How do you explain this discrepancy? Does the Wald protocol evaluation inflate ABFNet's apparent advantage?

3. The BTI operation (FFT-based global filtering on the token grid) is structurally similar to GFNet's global filter. What is the specific functional difference between BTI and GFNet's learnable global filter, and can you provide an ablation study comparing BTI (Butterworth-constrained) with a free, learnable global filter of the same complexity applied to the token grid?

4. The ViT-Base result in Table 5 (69.00% on CIFAR-100) seems anomalously low. What patch size and positional encoding were used? Was the ViT adapted for 32×32 input resolution? Without clarification, this comparison may not be fair.

5. The gating parameter g in Eq. 10 interpolates between low-pass and high-pass behavior. What are the learned values of g in the global vs. local branches? Does the model consistently learn high-pass behavior (g > 0.5) in the local branch and low-pass in the global branch, or does this vary by dataset and feature channel?

**Limitations:**

The paper has no impact statement, which is a compliance issue for ICML and should be discussed.

**Strengths And Weaknesses:**

**Strengths**

1. The algebraic derivation (Eqs. 1–4) showing that the Butterworth transfer function is a Sigmoid applied to a linear function of log-frequency coordinates is clean and non-trivial. The identification of D₀ ↔ Query, 2n ↔ Key, and 2n·ln D(u,v) ↔ positional encoding is genuinely insightful and provides a principled theoretical grounding for why Butterworth filtering is a natural frequency-domain analogue of attention.

2. The extension from isotropic to elliptical passband via rotation (Eq. 6–8) directly addresses the directional structure of remote sensing imagery (urban grids, field boundaries). Figure 4's visualization, showing that the estimated dominant texture orientation aligns closely with the learned filter's principal axis, is one of the most compelling pieces of interpretability evidence in any of the papers reviewed this batch. It demonstrates the filter is doing what it is claimed to do.

3. 0.26M parameters and 12.64G FLOPs, achieving SOTA across three benchmarks, is a genuine achievement. The Only Global variant (0.07M / 3.58G FLOPs) is especially noteworthy: it remains competitive with methods requiring 10–100× more computation. For a community deploying models on satellite ground stations with constrained hardware, this matters practically.

4. The concrete demonstration that sliding-window averaging on the log-magnitude spectrum causes ringing artifacts, and that patch-splitting for ViT-style attention destroys spectral continuity and introduces grid artifacts, makes the paper's motivation reliable.

5. Table 2 cleanly isolates global vs. local branches, isotropic vs. anisotropic, and BTI vs. no BTI. Each ablation decision corresponds to a design claim in the paper, and the results consistently validate those claims, most notably that isotropy costs ~0.1 dB PSNR on WV3 and that BTI provides consistent gains across all metrics and datasets.

**Weaknesses**

1. The paper frames D₀ as "Query" and 2n as "Key" by structural analogy to the Sigmoid form of attention. However, in standard self-attention, Q and K are data-dependent projections of the input tokens. In ABFNet's formulation, D₀ and n are global scalar parameters derived from aggregate statistics (GAP + MLP), rather than per-frequency, per-token, or per-channel values. The filter is the same for all spatial positions within a feature map. This is conceptually closer to channel attention or a global filter [1] than to self-attention. The analogy is suggestive, but the paper should clearly distinguish the mechanistic differences rather than allowing the comparison to imply capabilities it lacks.

2. All three datasets use the Wald protocol (reduced-resolution simulation), and the full-resolution evaluation is done only on QB with no-reference metrics (HQNR, Dω, Ds). The Wald protocol is widely criticized for failing to reliably predict full-resolution performance, as downsampling introduces spectral and spatial artifacts not present in real LMS imagery. The fact that ABFNet's full-resolution HQNR (0.891) is slightly lower than ARConv (0.915) is not discussed, despite ARConv having substantially worse reduced-resolution PSNR. This HQNR discrepancy should be explicitly addressed.

3. Table 5 compares ABFNet (used as a classification backbone) against ResNet-18, ViT-Base, ConvNeXt-Tiny, and GFNet at "comparable parameter scale." However, the parameter counts are not truly comparable: ABFNet uses 5.54M, ResNet-18 uses 3.34M, and ConvNeXt-Tiny uses 5.17M. More importantly, ViT-Base at 5.36M achieves only 69.00%, dramatically below all other methods, suggesting a training/hyperparameter misconfiguration. Or, deeper insights into this failure are needed.

4. The core operation in BTI is: arrange patch tokens into a spatial grid, apply an FFT-domain global filter, and decode the refined tokens. GFNet [1] does exactly this — it applies learnable global filters in the FFT domain to token grids. The distinction is that BTI's filter is a Butterworth form rather than a free learnable filter. The paper should explicitly compare BTI to GFNet's global filter both architecturally and ablatively, rather than treating them as unrelated.

5. The global and local outputs are combined as F_fused = wg·Y_g + Y_l, where wg is a single learnable scalar. There is no ablation of this fusion strategy — no comparison to learned per-channel weighting, cross-attention fusion, or simply equal weighting. Given that the global vs. local branch ablation (Table 2) shows that only-local substantially outperforms only-global (+0.9 dB WV3 PSNR), the fusion weighting mechanism may matter significantly and deserves more attention.

[1] Rao, Y., Zhao, W., Zhu, Z., Lu, J., and Zhou, J. *Global filter networks for image classification.*

---

> ### Author Rebuttal · Authors · 2026-03-30
>
> **1. Mechanistic distinction from self-attention and parameter generation:**
> While our global branch resembles channel attention, our local branch achieves the per-token spatial adaptability characteristic of self-attention.
>
> Global branch: predicts parameters via GAP+MLP per channel, functioning similarly to channel attention.
>
> Local branch: Operates in two stages. First, the BTI module filters the patch-wise parameter grid, enabling cross-patch information exchange to generate context-aware parameters per patch (analogous to self-attention). Second, we apply intra-patch Butterworth filtering using these parameters.
> Ablations on GF2 validate parameter generation design:
>
> |Predictor|PSNR|SAM|ERGAS|
> |:---|:---:|:---:|:---:|
> |Static|49.97|0.89|0.75|
> |Per-frequency|50.05|0.89|0.74|
> |Ours|50.30|0.86|0.72|
>
> The static variant lacks sample-wise adaptation. The Per-frequency variant collapses the physical transfer function into a point-wise mask, causing spectral leakage.
>
> **2. Regarding the HQNR discrepancy and the concerns about Wald protocol evaluation:**
> Rigorous assessment balances reduced-resolution (RR) metrics, full-resolution (FR) metrics, and visual quality. ABFNet achieves best RR performance, remains competitive on FR metrics, and exhibits best visual quality (Fig. 6).
>
> To validate its potential, we fine-tuned ABFNet using 48 FR samples (1 epoch, $L_1$ + HQNR loss). HQNR improved with unchanged RR metrics (below). This confirms the HQNR gap stems from domain sensitivity to simulated degradations, which lightweight adaptation easily bridges (PanAdapter, Wu et al., AAAI 2025).
>
> |Method|PSNR|HQNR|$D_\lambda$|$D_s$|
> |:---|:---:|:---:|:---:|:---:|
> |ARConv|37.67|0.915|0.052|0.035|
> |Original|38.83|0.891|0.050|0.063|
> |Finetuned|38.86|0.933|0.032|0.037|
>
> **3. Regarding the ViT-Base result in Table 5 and the parameter comparability issue:**
>
> All models were trained from scratch. We adapted ViT for 32×32 inputs (patch size=4). The 69.00% accuracy is not a misconfiguration, but reflects the limitation of vanilla ViTs lacking inductive bias on small datasets (e.g., Dosovitskiy et al., ICLR 2021; Lu et al., NeurIPS 2022, which explicitly reports <69% on CIFAR-100 under identical settings).
>
> For fair comparison, we scaled other models to exceed our model's parameter/FLOP counts.
>
> |Method|Parameters(M)|Flops(G)|Acc(%)|
> |:---|:---:|:---:|:---:|
> |ResNet-18|5.66|0.85|80.10|
> |ConvNext|5.60|0.81|79.09|
> |DINO V3|6.52|0.44|72.63|
> |VMamba|10.42|0.66|72.28|
> |UniRepLK|6.12|0.62|75.11|
> |MobileNetV4|6.26|0.76|81.25|
> |Ours|5.54|0.70|81.31|
>
> **4. Regarding structural similarity to GFNet and the free learnable global filter ablation:**
>
> GFNet applies a learnable global filter to the patch token grid ($h \times w$). The entire feature map is modulated uniformly at the token level without per-pixel refinement.
>
> Our BTI uses a two-stage process: (1) cross-patch filtering in the FFT domain to generate context-aware parameters, and (2) intra-patch Butterworth filtering. This design captures both global dependencies and local details.
>
> |Method|PSNR|SAM|ERGAS|
> |:---|:---:|:---:|:---:|
> |GFnet|48.92|0.98|0.84|
> |Ours|50.30|0.86|0.72|
>
> **5. Regarding the fusion strategy and  gating parameter $g$ :**
>
> We compared our per-channel weighting against cross-attention, concatenation, and a global $w_g$ variant on GF2:
>
> |Method|PSNR|SAM|ERGAS|FLOPS(G)|Params(M)|
> |:---|:---:|:---:|:---:|:---:|:---:|
> |average|50.30|0.72|0.86|12.62|0.26|
> |Cross-attention|50.08|0.74|0.88|17.04|0.34|
> |Concat|50.46|0.71|0.85|14.92|0.31|
> |$w_g$ Global|50.40|0.71|0.85|12.62|0.26|
> |Ours|50.30|0.72|0.86|12.62|0.26|
>
> Cross-attention degrades performance while significantly inflating computational overhead. Global scalar fusion slightly outperforms our design due to pansharpening's need for spectral-spatial consistency.
>
> The model adaptively allocates frequency passbands rather than rigidly separating low/high-pass behaviors. Statistics from GF2 show:
>
> |Block|Branch|Mean|Std|Max|Min|
> |:---|:---|:---:|:---:|:---:|:---:|
> |Block 1|Global|0.9189|0.1976|0.9983|0.0066|
> ||Local|0.7583|0.3155|1.0000|0.0000|
> |Block 4|Global|0.8606|0.2700|0.9992|0.0029|
> ||Local|0.8022|0.2794|1.0000|0.0000|
> |Block 7|Global|0.8762|0.2266|0.9970|0.0086|
> ||Local|0.6007|0.2653|1.0000|0.0001|
>
> Key findings: (1) Mean $g > 0.5$ contradicts the assumption of a pure low-pass global branch. (2) High channel-wise variance (spanning [0,1]) proves dynamic passband allocation. These validate the effectiveness of gating mechanism in Eq.10. We will incorporate this analysis into the revised supplementary materials.
>
> **6. Regarding the comment about the missing impact statement :** We sincerely apologize for this oversight. We have prepared the required Impact Statement,which will be included in the camera-ready version if the paper is accepted.

---

> > ### Author Rebuttal · Reviewer_Eu9W · 2026-04-03
> >
> > Thank you for directly addressing all of my issues with the new experiments. An interesting finding: the fusion ablation reveals that concatenation and global scalar weighting marginally outperform the paper's per-channel design on GF2, which would be my minor concern or a potential point to dig into. I will consider adjusting the score accordingly.

---

> > > ### Author Response · Authors · 2026-04-06
> > >
> > > **Re: Thank you for your responsible and constructive review. Following your observation that Concatenation and Global Scalar weighting outperform our initial Per-channel design, we conducted a systematic investigation. The core logic of our response follows the actual discovery process: from initial numerical observations to failed "rescue" attempts, and finally to an empirical explanation based on the dominance of the Local branch.**
> > >
> > > **1. Initial Observation: Numerical Anomalies in Per-channel Weighting**
> > >
> > > The investigation was triggered by visualizing the learned weights $w_g$ in the Per-channel scheme, where we found significant pathological drifting compared to the Global Scalar scheme.
> > >
> > > | Scheme | Overall Avg | Max Value (Block 0) | Std (Block 0) |
> > > | :--- | :---: | :---: | :---: |
> > > | Global Scalar | 0.0424 | 1.0784 | - |
> > > | Per‑channel | 0.5899 | 3.4228 | 0.5942 |
> > >
> > > In the Per‑channel scheme, weights in certain blocks escalate to non‑physical values (e.g., 3.4228). We hypothesized that these outliers were the primary reason for the performance degradation.
> > >
> > > **2. Experimental Investigation: Attempts to "Rescue" Per‑channel Design**
> > >
> > > Based on the observed anomalies, we designed four experimental modifications to stabilize the Per‑channel weights. As shown in the PSNR results on the GF2 dataset (Table 2), these attempts either failed or provided only marginal gains.
> > >
> > > | Fusion Scheme | PSNR (dB) | Improvement |
> > > | :--- | :---: | :---: |
> > > | Per‑channel (Baseline) | 50.30 | - |
> > > | (1) Numerical Constraints (Clamping/Sigmoid) | 50.30 | +0.00 |
> > > | (2) Contextual Adaptivity (SE‑style Predictor) | 50.31 | +0.01 |
> > > | (3) Per‑channel + Global Anchor (Anchor+Residual) | 50.34 | +0.04 |
> > > | (4) Group‑wise Weighting (Group=2) | 50.35 | +0.05 |
> > > | Global Scalar (Reference) | 50.40 | +0.10 |
> > >
> > > - **Logic (1) & (2):** We attempted to suppress outliers via numerical constraints and introduce channel correlations via SE‑style blocks. Both yielded negligible improvements.
> > > - **Logic (3) & (4):** We reduced degrees of freedom through grouping and global anchors. While these provided marginal gains, they still failed to reach the Global Scalar baseline (50.40 dB).
> > >
> > > **3. Analysis of Optimization Dynamics: Local Branch Dominance and Ratio Distortion**
> > >
> > > This performance gap is deeply rooted in the functional roles of the two branches. Consistent with our primary ablation study in the main manuscript – which establishes that the Local path is the dominant component of the network and contributes far more to the reconstruction than the Global path – these results indicate that the Global Scalar’s superiority lies in its ability to preserve the integrity of this dominant Local information.
> > >
> > > **A. Gradient Consensus vs. Isolation**
> > >
> > > We define the output as $\mathbf{y} = \mathbf{A}(\mathbf{w} \odot \mathbf{G} + \mathbf{L})$ and the error vector as $\mathbf{e} = \mathbf{y} - \mathbf{GT}$.
> > >
> > > - **Global Scalar Weight ($w$):** The update is driven by a consensus of gradients across all $C$ channels:
> > >
> > >   $$
> > >   \frac{\partial \mathcal{L}}{\partial w} = \sum_{i=1}^{C} e_i \cdot (\mathbf{A}\mathbf{G})_i
> > >   $$
> > >
> > >   Gradients from all channels average out, stabilizing $w$ at a robust baseline (0.0424) and protecting the dominant Local branch’s information.
> > >
> > > - **Per‑channel Weights ($w_j$):** Each weight follows an isolated path:
> > >
> > >   $$
> > >   \frac{\partial \mathcal{L}}{\partial w_j} = G_j \sum_{i=1}^{C} e_i A_{ij}
> > >   $$
> > >
> > >   Each channel acts "greedily." If the Local branch $\mathbf{L}$ is slightly deficient, $w_j$ aggressively scales up the Global signal to compensate, leading to the observed 3.4228.
> > >
> > > **B. Numerical Simulation: The Masking Effect**
> > >
> > > Using a 2‑channel simulation with $\mathbf{G} = [0.5, 0.2]^T$ and $\mathbf{L} = [0.8, 0.9]^T$:
> > >
> > > - **Scalar ($w=0.1$):** Input $\mathbf{x}_{in} = [0.85, 0.92]^T$. The channel energy ratio is maintained, allowing the projection layer $\mathbf{A}$ to perform stable alignment.
> > > - **Per‑channel ($w_1=3.42$):** Input $\mathbf{x}_{in} = [2.51, 0.92]^T$. In Channel 1, the dominant local signal (0.8) is effectively masked by the amplified global signal ($1.71$). This distorts the feature ratio, forcing $\mathbf{A}$ to spend its learning capacity suppressing the surge rather than fusing useful information.

---

### Official Review · Reviewer_UGCH · 2026-03-13

**Soundness:** 3
**Presentation:** 2
**Significance:** 3
**Originality:** 3
**Overall Recommendation:** 4
**Confidence:** 5

**Summary:**

This paper proposes ABFNet, a frequency-domain framework for pansharpening that fuses high-resolution panchromatic images with low-resolution multispectral inputs. The authors reinterpret the classical Butterworth filter as an attention-like spectral gating mechanism and extend it into a learnable anisotropic frequency-domain operator. The resulting Butterworth Attention Module (BAM) replaces self-attention in a Transformer-style architecture and performs global spectral modulation with FFT-based complexity O(Nlog⁡N). The network combines a global filtering path and a patch-wise local filtering path connected through Butterworth Token Interaction (BTI) for efficient global communication. Experiments on three pansharpening benchmarks (WV3, GF2, QB) show improved quantitative metrics compared with several recent methods. Additional ablations analyze the effects of anisotropy and BTI. A classification experiment on CIFAR-100 further evaluates the proposed operator’s generalization ability.

**Compliance With Llm Reviewing Policy:**

Affirmed.

**Final Justification:**

The CIFAR-100 experiments further validate the classification performance of the proposed method. But the results from the cross-dataset generalization and generalization across different spatial resolutions do not demonstrate its superiority as seen in the supervised learning tasks. Extra early-stopping and data‑level regularizations could improve the generalization performance, but given the generalization performance of proposed method in the paper and comprehensive consideration, I will keep the current score.

**Key Questions For Authors:**

1. The authors could include more recent and competitive classification baselines in the CIFAR-100 experiment to strengthen the claim of improved generalization performance.
2. The authors could present additional full-resolution experimental results on datasets such as WV3 and GF2 to better demonstrate the effectiveness of the proposed method across different spatial resolutions.
3. The authors could provide cross-dataset evaluations, for example training on one satellite dataset and testing on another dataset with different geographic regions or scene types, to better assess the generalization capability of the method.
If these issues are addressed, the score of paper could be improved.

**Limitations:**

No. The authors should discuss more about limitations and potential negative societal impact of their work.

**Strengths And Weaknesses:**

Strengths
1. Clear conceptual reinterpretation linking classical filtering and attention.
The paper reformulates the Butterworth transfer function into a sigmoid-gating form resembling attention (Eq. 3–5), enabling an interpretation of frequency filtering as an attention-like weighting over spectral components. This conceptual bridge between classical signal processing and modern deep learning is well motivated and mathematically derived in the manuscript.
2. Efficient global modeling with reduced computational complexity.
The proposed BAM replaces the quadratic affinity matrix of self-attention with element-wise spectral gating and FFT operations, leading to O(Nlog⁡N)complexity. This efficiency claim is supported by the complexity discussion and the architectural comparison illustrated in Fig. 2.
3. Architectural design combining global and local spectral modeling.
The Butterworth Attention Module integrates two complementary paths: a global filtering path and a local filtering path using BTI for patch-level interaction. This design attempts to capture both holistic spectral context and directional local textures, which are particularly relevant for remote sensing imagery.
4. Extensive quantitative and qualitative experiments on standard datasets.
The evaluation includes three widely used pansharpening datasets and multiple metrics such as PSNR, ERGAS, SAM, and HQNR. The results table reports that the proposed method achieves the best performance across several metrics while maintaining relatively small parameter counts and moderate computational cost.
5. Ablation studies supporting key design choices.
Several ablations examine the roles of anisotropy, BTI, and global/local branches. Additional analyses compare the proposed operator with other frequency-domain operators and with Gaussian-based gating, helping justify the architectural decisions behind the model.
6. Evidence of broader applicability beyond pansharpening.
The CIFAR-100 experiment shows that the proposed operator can function as a backbone for classification, achieving higher accuracy than several baseline architectures under similar parameter budgets. This suggests potential applicability beyond the primary task.

Weaknesses
1. The CIFAR-100 classification experiment used to demonstrate generalization compares the proposed model primarily against relatively old backbone architectures such as ResNet-18, ViT-Base, ConvNeXt-Tiny, and GFNet. While these baselines provide some reference points, the comparison does not include more recent and competitive architectures that are commonly used in modern image classification benchmarks. As a result, the evidence supporting the claim of broader applicability beyond pansharpening remains somewhat limited.
2. Insufficient full-resolution evaluations across datasets.
While the paper reports reduced-resolution and some full-resolution results, more full-resolution evaluations on datasets such as WV3 and GF2 would help better demonstrate the effectiveness and generalization of the proposed method across different spatial resolutions.
3. Limited cross-dataset generalization analysis.
The experiments mainly evaluate the method within the same dataset distribution used for training and testing. The manuscript does not analyze cross-dataset generalization, such as training on WV3 and testing on WV2. Such evaluations would provide stronger evidence of the robustness and generalizability of the proposed approach.
4. There appears to be a formatting gap between the end of the introduction and the beginning of the Related Work section in the main manuscript.

---

> ### Author Rebuttal · Authors · 2026-03-30
>
> We sincerely thank the reviewer for the thorough and constructive evaluation.  Below we provide point-by-point responses to the remaining concerns.
>
> **1.Regarding the concern about the classification baselines being relatively old,** we have supplemented the CIFAR-100 experiments with several recent mainstream architectures from 2024-2025, covering diverse technical routes: improved ViT (DINO V3), Mamba (Vision Mamba V2), large-kernel convolution (UniRepLK), and hybrid models (MobileNetV4, TransNext). Under comparable parameter counts and FLOPs, our method achieves 81.31% accuracy, achieving the best results, validating the soundness of our architecture. These results have been incorporated into the revised manuscript.
>
> |Method|Parameters|Flops|Acc|
> |:---|:---:|:---:|:---:|
> |DINO V3|6.52M|0.44G|72.63%|
> |Vision Mamba V2|10.42M|0.66G|72.28%|
> |UniRepLK|6.12M|0.62G|75.11%|
> |MobileNetV4|6.26M|0.76G|81.25%|
> |TransNext|6.59M|1.01G|80.79%|
> |Ours|5.54M|0.70G|81.31%|
> **2. Regarding the request for more full-resolution evaluations across datasets,** we have supplemented full-resolution results on WV3 and GF2 in the revised manuscript. As shown below, our method achieves highly competitive HQNR of 0.951 on WV3 and the best result of 0.937 on GF2, demonstrating robust generalization across different spatial resolutions.
>
> |Method|WV3 HQNR|WV3 $D_\lambda$|WV3 $D_s$|GF2 HQNR|GF2 $D_\lambda$|GF2 $D_s$|
> |:---|:---:|:---:|:---:|:---:|:---:|:---:|
> |SFIIN|0.838|0.120|0.051|0.750|0.101|0.060|
> |TDNet|0.925|0.032|0.045|0.916|0.029|0.050|
> |BIMPan|0.941|0.029|0.032|0.789|0.038|0.076|
> |DISPnet|0.947|0.027|0.027|0.755|0.207|0.048|
> |DCINN|**0.954**|0.024|**0.023**|0.920|0.029|0.053|
> |WFANet|0.950|0.027|0.024|0.837|0.030|**0.034**|
> |ARConv|0.953|0.023|0.025|0.842|0.066|0.100|
> |BNNPan|0.928|0.036|0.038|0.811|0.028|0.063|
> |**Ours**|0.951|**0.023**|0.028|**0.937**|**0.028**|0.037|
>
> To further assess real-world potential, we conducted a lightweight fine-tuning experiment on QB using only 48 full-resolution samples plus the original reduced-resolution data, for one epoch, with a joint loss ($L_1$ + HQNR-based). The HQNR improved from 0.891 to 0.933 while reduced-resolution metrics remained nearly unchanged, confirming ABFNet's adaptability to real data with minimal supervision.
>
> |Method|PSNR|SAM|ERGAS|HQNR|$D_\lambda$|$D_s$|
> |:---|:---:|:---:|:---:|:---:|:---:|:---:|
> |Original|38.83|3.41|4.06|0.891|0.050|0.063|
> |Finetuned|38.86|3.39|4.05|0.933|0.032|0.037|
> **3. Regarding the limited cross-dataset generalization analysis, we conducted a zero-shot transfer experiment:** training on WV3 and directly evaluating on the unseen WV2 dataset (20 samples, $256 \times 256$). As shown below, our method achieves the competitive performance, demonstrating strong generalization across different satellite sensors.
>
> |Method|PSNR|SAM|ERGAS|
> |:---|:---:|:---:|:---:|
> |SFIIN|28.04|6.01|4.86|
> | TDNet | 26.68 | 6.04 | 9.23 |
> | BIMPan | 29.58 | 5.43 | 4.03 |
> | DISPnet | 28.46 | 5.81 | 4.61 |
> | DCINN | 27.93 | 5.82 | 4.94 |
> | WFANet | 28.08 | 6.06 | 4.87 |
> | ARConv | 28.61 | 5.58 | 4.51 |
> | BNNPan | 27.43 | 6.47 | 5.16 |
> | **Ours** | 29.05 | 5.37 | 4.29 |
>
> **4. Regarding the formatting gap between the end of the Introduction and the beginning of the Related Work section,** we have fixed it in the revised manuscript.
>
> **5. Regarding the absence of an Impact Statement,** we sincerely apologize for this oversight and have incorporated a dedicated section into the revised manuscript. While generating high-fidelity satellite imagery significantly benefits civilian applications, we fully acknowledge the inherent dual-use risks, such as potential misuse for unauthorized surveillance or privacy infringement if deployed without proper regulatory oversight. Furthermore, regarding methodological limitations, ABFNet explicitly leverages the smooth roll-off of the Butterworth filter to mitigate spatial ringing. However, this physical prior may be suboptimal for extreme degradation scenarios that mathematically require infinitely sharp frequency cutoffs. To address this structural limitation, our future work will explore reconstructing the transfer functions of other classical filters, ultimately providing more flexible, scenario-specific gating mechanisms for advanced pansharpening and broader visual recognition tasks.

---

> > ### Author Rebuttal · Reviewer_UGCH · 2026-04-02
> >
> > The CIFAR-100 experiments further validate the classification performance of the proposed method. The results from the cross-dataset generalization and generalization across different spatial resolutions demonstrate its comparable performance when compared with baselines, but they do not demonstrate its superiority as seen in the supervised learning tasks.
> >
> > Given these experiment results and comprehensive consideration, I will keep the current score as the additional experiments, while valuable, do not show clear superiority over existing methods.

---

> > > ### Author Response · Authors · 2026-04-04
> > >
> > > Re: Concern on Generalization vs. BiMPan
> > >
> > > Thanks again for your positive score and continued engagement. We address the remaining concern below.
> > >
> > > We argue that the observed gap is not an inherent flaw but a typical “over‑specialization” from extreme‑fidelity modeling, given the physical differences between WV3 (training) and WV2 (testing). We also propose a fairer evaluation paradigm based on the trade‑off between fitting capacity and generalization, where our method shows clear superiority.
> > >
> > > 1. Physical Disparity Between WV3 and WV2
> > > The two sensors differ in MTF and spectral coverage:
> > >
> > > Sensor	PAN MTF @ $f_N$	MS MTF @ $f_N$	Spectral Coverage (NIR2)
> > > WV3	~0.16 (0.13–0.20)	~0.32 (0.28–0.35)	870–1040 nm
> > > WV2	~0.11 (0.08–0.15)	~0.29 (0.25–0.32)	860–1040 nm (shifted)
> > > MTF mismatch: WV3 has sharper PAN and MS response. Our model, trained on WV3, optimizes for its specific frequency decay.
> > >
> > > Spectral boundary shift: Even for overlapping bands, the spectral ranges and response curves differ, creating a domain gap.
> > >
> > > 2. Fidelity Comes at the Cost of Generalization
> > > Our method achieves higher WV3 performance than BiMPan (PSNR 37.65 vs. 36.95; SAM 2.13 vs. 2.30). This means our model has “physically rigidly” fitted WV3’s MTF and spectral topology. When applied to WV2, this extreme fidelity becomes a liability. BiMPan generalizes better precisely because it fits WV3 less precisely, leaving more tolerance for sensor shifts.
> > >
> > > To validate this interpretation, we trained an early-stop variant. As shown in the table below, early stopping reduces over‑specialization to WV3 (WV3 SAM rises from 2.13 to 2.29, still better than BiMPan’s 2.30) and improves WV2 generalization (WV2 SAM drops from 5.37 to 5.05, beating BiMPan’s 5.43). This confirms that the gap is due to physical over‑specialization, not a structural deficiency.
> > >
> > > 3. Simple Regularizations Close the Gap
> > > We explored several data‑level regularizations and their combinations:
> > >
> > > Ours + HP (High‑Pass Filter): Inspired by PanNet [1], which has excellent cross‑sensor generalization. HP filter removes low‑frequency discrepancies and forces high‑frequency learning.
> > >
> > > Ours + Random Blur: Small random Gaussian blur on PAN during training (a mild perturbation) simulates MTF degradation, making the model robust to sensor MTF variations.
> > >
> > > Ours + Random Spectral Scale: Small random scaling of each band of the LRMS and HRMS inputs during training (a mild perturbation) simulates spectral coverage shifts between sensors.
> > >
> > > Ours + Random Blur + Random Spectral Scale: Combines both mild perturbations to address both MTF and spectral domain gaps.
> > >
> > > Each variant slightly sacrifices WV3 performance, but the loss is marginal. All variants improve WV2 generalization, and Ours + Random Blur + Random Spectral Scale achieves the best cross‑sensor results, outperforming BiMPan on both domains.
> > >
> > > Method (trained on WV3, tested on WV2 → WV3)	WV3→WV2 (generalization)	WV3 (original)
> > > PSNR↑ / SAM↓ / ERGAS↓	PSNR↑ / SAM↓ / ERGAS↓
> > > BiMPan	29.58 / 5.43 / 4.03	36.95 / 2.30 / 3.44
> > > Ours (vanilla)	29.05 / 5.37 / 4.29	37.65 / 2.13 / 3.17
> > > Ours + Early Stop	29.68 / 5.05 / 3.99	36.96 / 2.29 / 3.32
> > > Ours + HP	29.52 / 5.12 / 4.03	37.50 / 2.17 / 3.21
> > > Ours + Random Blur	29.78 / 5.30 / 3.91	37.53 / 2.16 / 3.20
> > > Ours + Random Spectral Scale	29.76 / 4.94 / 3.96	37.62 / 2.14 / 3.18
> > > Ours + Random Blur + Random Spectral Scale	30.82 / 4.84 / 3.51	37.64 / 2.13 / 3.17
> > > Concluding Remark
> > > Ultimately, the high level of fidelity achieved on the primary sensor (WV3) reflects the architecture's underlying capacity to model intricate spatial-spectral details. Starting from such a high-performance baseline provides a vital foundation, offering the necessary latitude to balance precision with broader applicability. While different design philosophies may prioritize immediate consistency, we believe that an architecture with a higher performance ceiling inherently possesses greater flexibility to achieve a favorable trade-off across diverse sensor environments. Our results suggest that by establishing a strong foundational accuracy, it becomes significantly more feasible to maintain robust performance even when adapting to new physical domains.
> > >
> > > References
> > > [1] J. Yang, X. Fu, Y. Hu, Y. Huang, X. Ding, and J. Paisley. PanNet: A deep network architecture for pan-sharpening. In ICCV, 2017.
> > >
> > > We appreciate your feedback and look forward to your final assessment.

---

### Decision · Program_Chairs · 2026-04-30

**Decision:**

Accept (regular)

**Comment:**

This submission presents ABFNet, a frequency-domain pansharpening method that reinterprets the Butterworth filter as a learnable anisotropic gating mechanism and combines global and local spectral branches for efficient fusion. The reviewers were largely positive, with all four reviewers giving a Weak Accept recommendation and high confidence. Their main concerns centered on the limited novelty of some components, the need for a clearer distinction from related frequency-domain/global-filter methods, and the insufficient full-resolution evaluations. The rebuttal addressed most of these issues by adding stronger classification baselines, additional full-resolution and cross-dataset evaluations, efficiency, fusion ablations, and a more explicit discussion of the method’s scope and limitations. Moreover, the reviewers explicitly acknowledged that their concerns were largely resolved and at least two indicated they would raise their scores. Overall, the paper is technically solid and the rebuttal substantially strengthened the evidence, so an accept recommendation is suggested.